# Harnessing protein language model for structure-based discovery of highly efficient and robust PET hydrolases

Banghao Wu [1,2,3,6], Bozitao Zhong [1,2,6], Lirong Zheng [2,5] ✉, Runye Huang[1,2,3], Shifeng Jiang[1], Mingchen Li [2,4], Liang Hong [1,2,3,4] ✉ & Pan Tan [1,2,4] ✉

Plastic waste, particularly polyethylene terephthalate (PET), presents significant environmental challenges, driving extensive research into enzymatic biodegradation. However, existing PET hydrolases (PETases) are limited by narrow sequence diversity and suboptimal performance. This study introduces VenusMine, a protein discovery pipeline that integrates protein language models (PLMs) with a representation tree to identify PETases based on structural similarity using sequence information. Using the crystal structure of *Is*PETase as a template, VenusMine identifies and clusters target proteins. Candidates are further screened using PLM-based assessments of solubility and thermostability, leading to the selection of 34 proteins for biochemical validation. Results reveal that 14 candidates exhibit PET degradation activity across 30–60 °C. Notably, a PET hydrolase from *Kibdelosporangium banguiense* (*Kb*PETase) demonstrates a melting temperature ($T_m$) 32 °C higher than *Is*PETase and exhibits the highest PET degradation activity within 30 – 65 °C among wild-type PETases. *Kb*PETase also surpasses FastPETase and LCC in catalytic efficiency. X-ray crystallography and molecular dynamics simulations show that *Kb*PETase possesses a conserved catalytic domain and enhanced intramolecular interactions, underpinning its improved functionality and thermostability. This work demonstrates a novel deep learning approach for discovering natural PETases with enhanced properties.

Plastic waste poses serious risks to human health and the environment, contributing to pollution, toxic chemical exposure, and ecosystem disruption[1–3]. Recycling plastic is therefore essential to mitigate these negative impacts, conserve natural resources, and reduce environmental contamination. Polyethylene terephthalate (PET), one of the most widely used plastics, has already been extensively utilized in packaging, textiles, and various consumer products[4]. Due to its durability and extensive use, PET accumulation in landfills and natural ecosystems has become a significant environmental issue[1,2,5]. Traditional mechanical and chemical recycling methods for PET are limited in both efficiency and environmental sustainability[6–8]. As a result, developing biological solutions for PET degradation, particularly through the use of hydrolase enzymes to product high-quality recycled PET (rPET), has obtained significant attention from both scientific research and industry[9–18].

[1]School of Life Sciences and Biotechnology, Shanghai Jiao Tong University, Shanghai, China. [2]Shanghai National Center for Applied Mathematics (SJTU Center) & Institute of Natural Sciences, Shanghai Jiao Tong University, Shanghai, China. [3]Zhang Jiang Institute for Advanced Study, Shanghai Jiao Tong University, Shanghai, China. [4]Shanghai Artificial Intelligence Laboratory, Shanghai, China. [5]Present address: Department of Cell and Developmental Biology & Michigan Neuroscience Institute, University of Michigan Medical School, Ann Arbor, Michigan, USA. [6]These authors contributed equally: Banghao Wu, Bozitao Zhong. ✉e-mail: lrzheng@umich.edu; hongl3liang@sjtu.edu.cn; tpan1039@gmail.com

Multiple enzymes capable of degrading PET have been biochemically and structurally identified and characterized[19–23]. The most effective PET-degrading enzymes to date are derived from culturable microorganisms isolated from environmental microbial communities, such as the widely used *Ideonella sakaiensis* PET hydrolase (*Is*PETase)[12]. However, approximately 99% of microbial species in nature are uncluturable, making metagenomic data a crucial resource for discovering new enzymes[24]. Successes in this field include the identification of leaf and branch compost cutinase (LCC)[19], PHL-7 (PES-H1)[22,25], PE-H[26], Bacterium HR29 (*Bhr*PETase)[21], and *Thermobifida fusca* cutinases (*Tf*Cut1 and *Tf*Cut2)[20]. Recently, a thermophilic PET-degrading enzyme, MG8, was discovered from human saliva metagenomic data[23]. Despite these advancements, the performance of these wild-type (WT) PETases remains insufficient for industrial PET enzymatic hydrolysis. Industrial-scale PET biodegradation requires high-temperature conditions to achieve high yields of rPET[27,28], as temperatures approaching PET's glass transition release molecular energy, facilitating PETase-catalyzed hydrolysis[18,29–34]. However, WT PETases are limited by low catalytic activity and thermostability[10,31,35]. Therefore, discovering new WT PETase with high catalytic activity and thermostability are crucial for advancing PETase engineering for the PET biodegradation industry.

The discovery of functional proteins is crucial for advancing both biotechnology and the life sciences[36]. Sequence-based screening remains the most widely used approach for enzyme discovery[36,37], leading to the identification of enzymes such as LCC[19] and MG8[23]. Although sequence-based methods-such as identifying conserved residues, sequence similarities, or hidden Markov models (HMMs)-are effective, proteins with similar functions may still go undetected when rely solely on sequence information[38]. Conversely, because the three-dimensional (3D) structure of a protein largely dictates its function, structure-based discovery offers a more robust approach, yielding a more diverse repertoire for enzyme mining[39,40]. While structural data in public databases remains limited[41,42], deep learning methods, including AlphaFold[43], have shown great promise in accurately predicting protein 3D structures, enabling the large-scale exploration and classification of proteins with structure queries.

In this study, we develop VenusMine, a structure-based enzyme discovery pipeline integrating sequence/structure retrieval tools and protein language models to identify thermostable PETases. Through systematic screening of 34 selected candidates using VenusMine, we identify 14 active PETases, including 8 with a melting temperature 10 °C higher than *Is*PETase and 3 demonstrating significantly enhanced catalytic activity. Notably, PETase from *Kibdelosporangium banguiense* (*Kb*PETase) exhibits exceptional properties, with a $T_m$ 32 °C higher than *Is*PETase and catalytic activity at 50 °C that outperforms *Is*PETase and LCC under their optimal reaction conditions. *Kb*PETase also displays enhanced catalytic efficiency ($k_{cat}/K_M$) compared to FastPETase and LCC. Overall, we establish a structure-based enzyme mining approach combined with protein language models, which is both reliable and sensitive for identifying and screening candidate PETase, thereby accelerating the discovery of high-performance enzymes.

## Results

### VenusMine: A structure-base enzyme discovery pipeline
Our structure-based enzyme discovery pipeline for PETase consists of two primary stages (Fig. 1a). First, we aimed to compile a comprehensive collection of potential PETases sharing structural similarity with known enzymes. Starting with *Is*PETase as the query structure (PDB: 5XFY[44]), we utilized the structure search tool FoldSeek[45] to perform structure similarity searches (see details in Methods). Given that structure databases are still limited (e.g., AFDB and ESMAtlas), we used the identified proteins as queries to conduct an extensive sequence search through the NCBI NR sequence database using MMseqs2 (see details in Methods). This approach resulted in a vast repertoire of 33,247,501 candidates, which could cover almost all

potential PETases that share a similar structure to our query protein, *Is*PETase (Fig. 1b). This approach resulted in a vast repertoire of 33,247,501 candidates and kept 436,488 after 50% sequence identity clustering, which could cover almost all potential PETases that share a similar structure to our query protein, *Is*PETase (Fig. 1b).

The second stage of the pipeline clusters candidate proteins to identify those most likely to exhibit optimal functionality and desired characteristics. Here, we utilized the sequence embeddings from the protein language model ProstT5[46] to represent the proteins. To reduce computational cost, we used representative sequences from 50% identity sequence clusters to calculate embeddings for subsequent analysis. ProstT5 has learned a mapping from protein sequences to structures, effectively representing the proteins in the structure space. This means that proteins with closer embeddings in Euclidean space are more likely to share similar structures. Subsequently, agglomerative clustering was applied to the protein embeddings to group the candidate enzymes and build a representation tree (Fig. 1c, d). To elucidate the functions and characteristics of each cluster, a list of well-studied enzymes with annotations are clustered together for comparison. The agglomerative clusters enabled us to pinpoint the most promising clade for PET degradation activity, which contains more confirmed PETase, while others contained enzymes from different EC numbers (Fig. 1d).

For further screening, we selected two clusters (cluster 1 and 3), as these clusters contained all reported high-activity wild-type PETases (e.g., *Is*PETase, LCC, TFH). A full list of known PETases within these clusters is provided in Supplementary Data 2. These clusters underwent multi-tiered screening: (1) A fine-tuned ESM2[47,48] model predicted $T_m$, with sequences demonstrating $T_m$ larger than that of *Is*PETase retained; (2) ProtSolM[49] eliminated candidates with low solubility predictions; (3) ESMfold-predicted structures were aligned with *Is*PETase (PDB:5XFY) using TM-align, excluding proteins with TM-score below 0.5 or sequences exhibiting catalytic triad mismatches. Finally, the 34 top-ranked candidates by predicted $T_m$ were finally selected for experimental validation, balancing computational predictions with practical experimental throughput constraints (Supplementary Fig. 1). This hierarchical approach ensured systematic identification of thermostable, soluble candidates while preserving structural fidelity to functional PETase architectures.

### Characterizations of putative PET hydrolases
We analyzed the sequences of 34 candidates to identify signal peptides using SignalP, followed by truncation to enhance expression efficiency (Supplementary Figs. 2–3)[50,51]. After codon optimization (Supplementary Data 1), the modified sequences were cloned into the pET28a (+) vector and expressed in *E. coli* BL21(DE3) with an N-terminal His-tag for purification. Among the candidates, 26 were successfully expressed and purified. The high expression success rate can also be attributed to the effective pre-screening capabilities of the PLM in the initial selection process. The *p*-nitrophenyl butyrate (*p*NPB) molecule contains an ester bond, and PETase similarly catalyzes the cleavage of ester bonds in PET (Supplementary Fig. 4). Therefore, we employed *p*NPB as a substrate to rapidly assess ester bond hydrolysis activity by measuring the absorbance of the reaction product, a method widely utilized in PETase discovery and related research[12,18,19,33,52,53] (Supplementary Figs. 4 and 5). After incubating proteins with substrates at 37 °C, 14 of the 26 expressible proteins exhibited ester bond-cleaving activity (Fig. 2a).

To evaluate PET degradation performance[50], we used commercially available amorphous PET film (AF-PET, Goodfellow Cambridge Ltd, Cat. No. ES301445) as the substrate, which is widely used for assessing PET degradation activity[13,54]. The aromatic reaction products mono(2-hydroxyethyl) terephthalate (MHET) and terephthalic acid (TPA) were quantified using ultra-high-performance liquid chromatography (UPLC)[13]. For comparative analysis, we used *Is*PETase, the

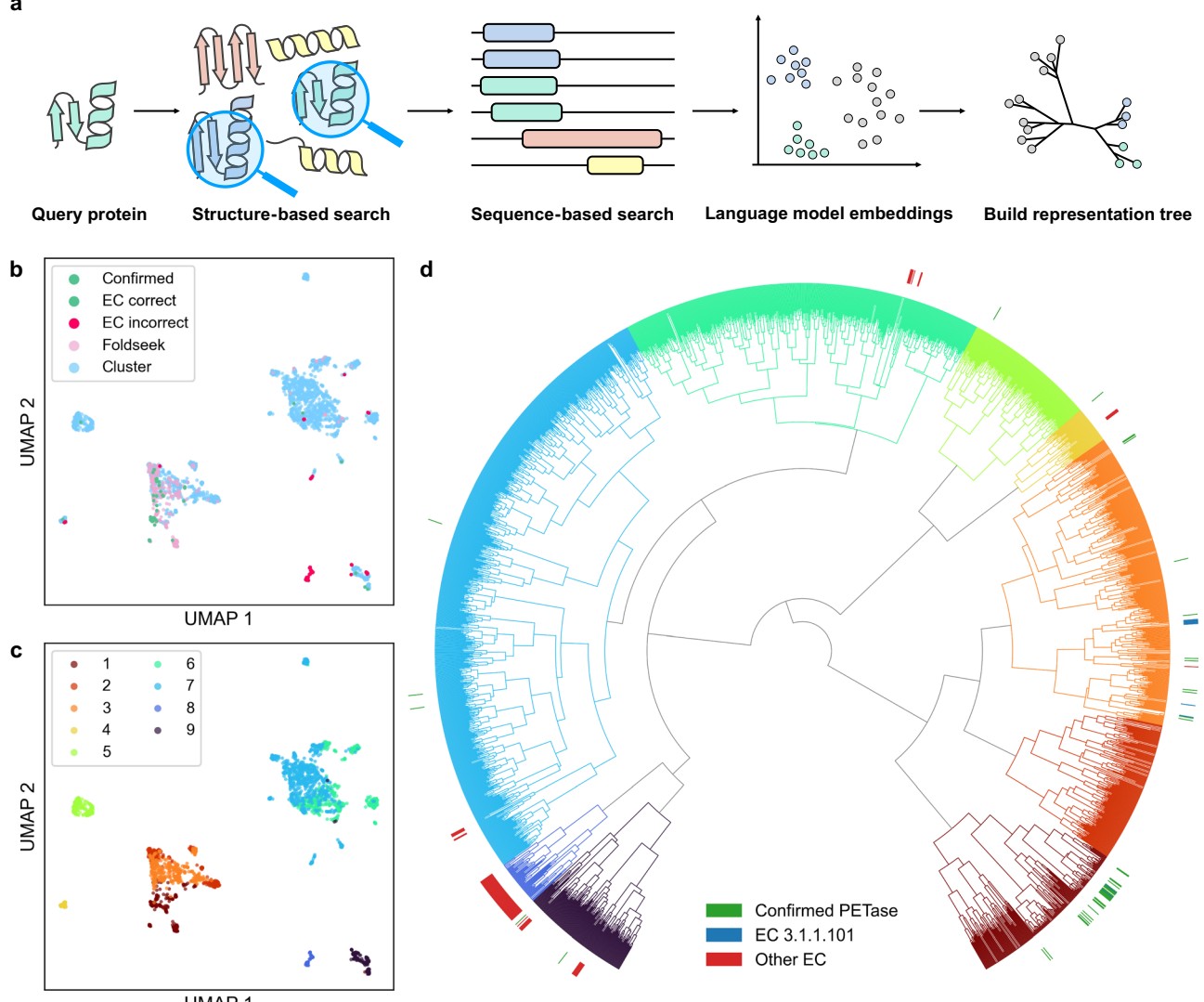

**Fig. 1 | In silico structure-based PETase discovery pipeline and clustering results. a** The pipeline for PETase discovery combines structure retrieval with FoldSeek, sequence retrieval with MMSeqs2, protein embeddings computation via ProstT5, and representation tree construction to identify functional and robust PETase. **b** The UMAP visualization of the PETase candidates identified by ProstT5. Green points indicate proteins with previously validated PET hydrolytic activity or annotated with corrected EC number 3.1.1.101, while red points represent those not annotated with the corresponding EC number. Pink points represent proteins

identified using FoldSeek, and orange points correspond to proteins discovered through sequence-based searches. **c** The UMAP visualization of the PETase candidates colored by representation tree clusters. **d** The representation tree constructed using embeddings from ProstT5. Green bars represent proteins with previously validated PETase activity, blue bars present enzymes with the correct EC number, while red bars indicate enzymes not annotated with the correct EC number. Two clusters are selected for further screening: the first cluster (dark red) and the third cluster (orange), arranged in a counterclockwise direction.

reference protein in our structural searches, as a control baseline. Since the optimal catalytic temperatures of the enzymes were unknown, we tested the activity of the 14 candidates across a range of 30−60 °C. All proteins exhibited activity within this range, with 11 of the 14 displaying degradation activity comparable to *Is*PETase (Fig. 2c). Given that $T_m$ is an indicator of PETase thermostability and performance[11], we used differential scanning fluorimetry (DSF) to determine the $T_m$ of the proteins. The results showed that the $T_m$ of the 14 proteins ranged from 36.4 °C to 80.1 °C (Fig. 2b), with eight candidates demonstrating superior thermostability compared to the *Is*PE-Tase, indicating the VenusMine's effectiveness in identifying thermostable and functional PETases. We also observed that the three aforementioned properties exhibited low correlations (Supplementary Fig. 6), highlighting the complexity of PETase activity and suggesting that multiple factors likely contribute to overall degradation performance. However, the PLM was able to capture relevant features, enabling high-positive screening of candidate PETases.

Notably, APET-14 showed the highest catalytic activity at 50 °C, with a 97-fold increase compared to *Is*PETase at 30 °C (Fig. 2c). and demonstrated excellent thermostability, with a $T_m$ increase of 32.4 °C over *Is*PETase, as validated by nanoDSC (Supplementary Fig. 7). These results indicate that APET-14 (GenBank: WP_209642273.1, PETase from *Kibdelosporangium banguiense*[55], hereafter *Kb*PETase) combines high enzymatic activity and thermostability, traits essential for efficient PET degradation, and was thus selected for further evaluation. In addition, the sequence similarity between *Kb*PETase and other known PETases ranges from 30% to 50%, highlighting the capability of VenusMine to identify functional PETase in regions of low sequence similarity (Supplementary Fig. 8).

### Identification of *Kb*PETase exhibiting high PET hydrolytic activity and thermostability

We compared the activity of *Kb*PETase to that of other characterized PET degrading enzymes, such as *Is*PETase[12] (mesophilic), PE-H[26]

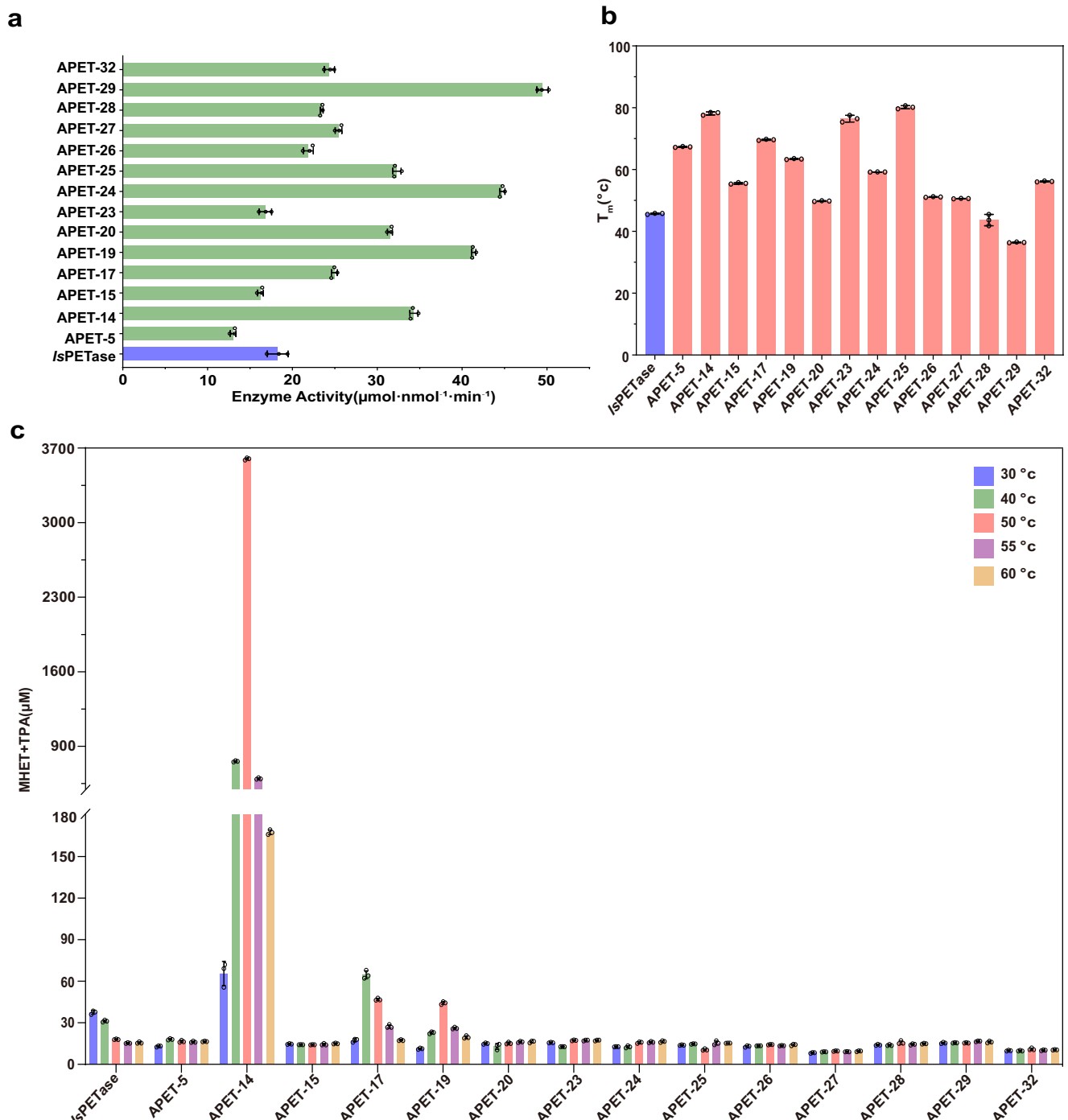

**Fig. 2 | Experimental validation and biochemical characterization of candidates. a** Enzyme activity of 14 proteins showing effective ester bond hydrolysis, as measured using *p*-nitrophenyl butyrate (*p*NPB) as a substrate at 37 °C. Reactions were performed in triplicate; data are presented as mean values ± SD. **b** The $T_m$ of the 14 proteins exhibiting ester bond cleavage activity. Reactions were performed in triplicate; data are presented as mean values ± SD. **c** Comparison of the degradation activity of candidate proteins on PET films. The reaction was conducted in 50 mM Glycine-NaOH (pH 9.0) at various temperatures (30, 40, 50, 55, and 60 °C) for 72 h. Reactions were performed in triplicate; data are presented as mean values ± SD.

(mesophilic), BTA-2[56] (mesophilic), Cut_190[53] (thermophilic), Thc_Cut1[57] (thermophilic), TFH[58] (thermophilic), and LCC[19] (thermophilic), across a temperature range of 30–65 °C (Fig. 3a). Note that Cut190, *Is*PETase, LCC, and PE-H are used as superior starting templates for directed evolution in industry[18].

Compared to *Is*PETase, which exhibits the highest catalytic activity at room temperature moderate temperatures with its optimal catalytic activity at 30 °C[12], *Kb*PETase demonstrates a 97-fold higher catalytic activity at its optimal temperature of 50 °C. (Fig. 3a). To assess

thermostability, we analyzed the $T_m$ of *Kb*PETase and other enzymes, finding that its enhanced activity likely correlates with its superior thermostability (Fig. 3b). Time-dependent PET degradation further revealed the stability of *Kb*PETase, with continuous activity enhancement over time, unlike *Is*PETase, which plateaued after 24 h at 30 °C (Supplementary Fig. 9). Compared to LCC, a highly active thermophilic PETase[19], *Kb*PETase exhibits 5.7-fold higher activity under the same reaction conditions at 50 °C, and even outperformed LCC by 1.47-fold at its optimal temperature of 65 °C (Fig. 3a) (Fig. 3a), despite the fact

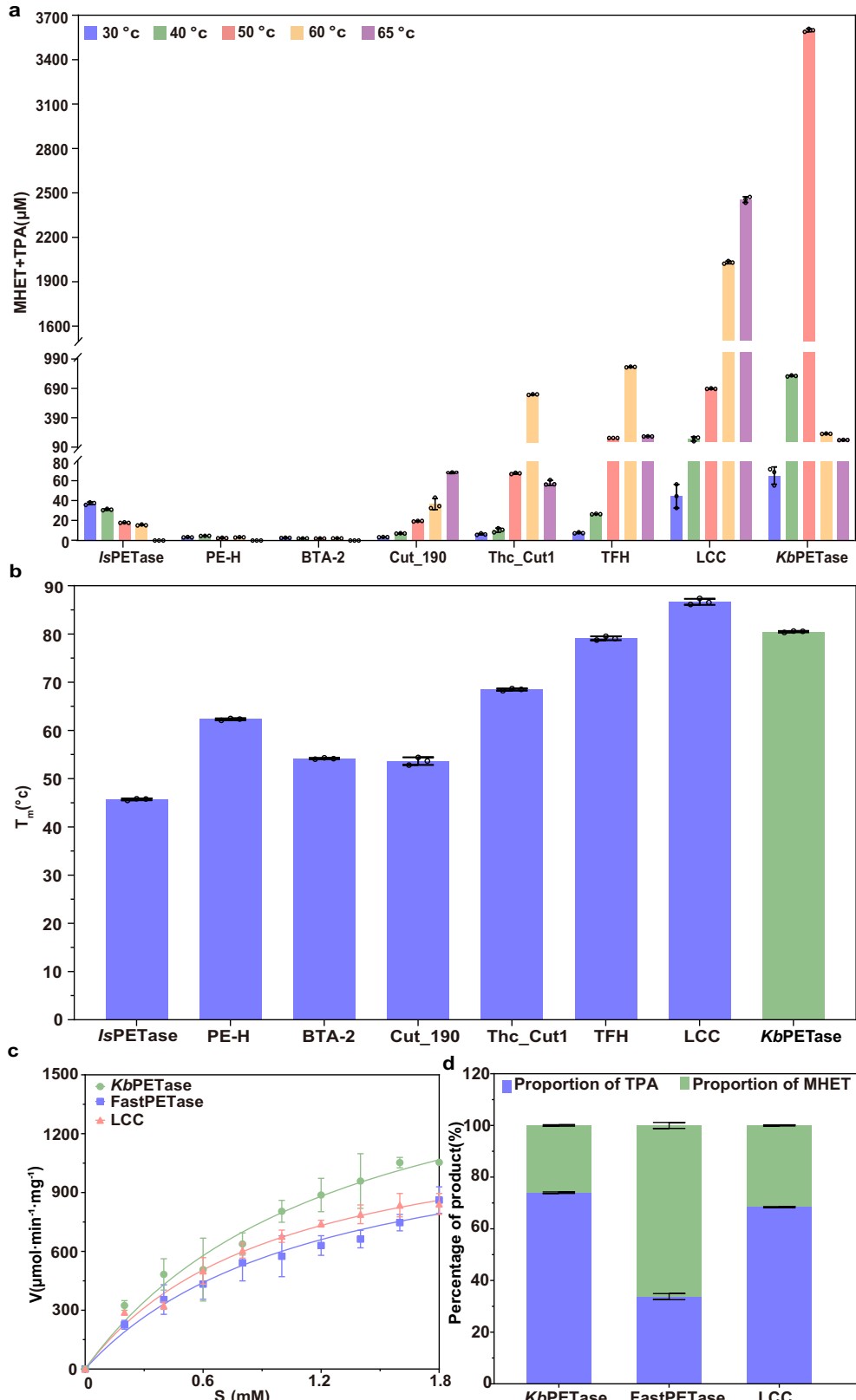

**Fig. 3 | Activity and thermostability measurements of discovered PETases.**
**a** PET film degradation activity of the previously characterized PETases compared to *Kb*PETase. The reaction was conducted in 50 mM Glycine-NaOH (pH 9.0) at various temperatures (30, 40, 50, 55, 60 and 65 °C) for 72 h. Reactions were performed in triplicate; data are presented as mean values ± SD. **b** Comparison of the $T_m$ of *Kb*PETase with other reported WT PETases using DSF. Reactions were performed in triplicate; data are presented as mean values ± SD. **c** Comparison of the

enzymatic kinetics curves of *Kb*PETase, LCC and FastPETase using *p*NPB as the substrate. Reactions were performed in triplicate; data are presented as mean values ± SD. **d** The depolymerization percentages of PET by *Kb*PETase and FastPETase at 50 °C, as well as by LCC at 65 °C, were determined through UPLC analysis of the released products. Reactions were conducted in triplicate, and the results are presented as mean values ± SD.

that higher temperatures closer to PET glass transition temperature can facilitate faster degradation[7,18,27,32]. This superiority persisted when LCC was tested in its optimal 100 mM potassium phosphate buffer (pH 8.0) at 65 °C, with *Kb*PETase at 50 °C maintaining 1.15-fold higher activity (Supplementary Fig. 10). In addition, the degradation products of *Kb*PETase exhibited a slightly higher proportion of TPA (Fig. 3d), a critical factor for PET recycling[27,59]. However, when we tested the Michaelis-Menten constants[33], we found that while the $K_m$ values were similar, *Kb*PETase exhibited a 1.5-fold higher $k_{cat}$ and a 1.3-fold higher catalytic efficiency ($k_{cat}/K_M$) compared to LCC (Fig. 3c and Table 1). These findings indicate that *Kb*PETase combines the moderate-temperature catalytic efficiency typical of mesophilic enzymes with the elevated activity characteristic of thermophilic counterparts.

We also compared the performance of *Kb*PETase with FastPETase, the most active mutant derived from *Is*PETase, known for its optimal catalytic performance also at 50 °C[54]. *Kb*PETase exhibits a $T_m$ that is 8 °C higher than that of FastPETase. Evaluation of PET degradation activity across a temperature gradient revealed that *Kb*PETase demonstrated 1.71-fold higher activity than FastPETase at their shared optimal temperature (Supplementary Fig. 11). To investigate the underlying reasons

for this difference, we also determined the Michaelis-Menten kinetic constants of both enzymes using *p*NPB as a substrate[33]. *Kb*PETase demonstrates a 1.87-fold higher catalytic efficiency compared to FastPETase, along with superior substrate affinity (Fig. 3c and Table 1). Besides, *Kb*PETase produced TPA at a concentration that is 2.2-fold higher than that of FastPETase (Fig. 3d). However, when compared to the engineered variant LCC[ICCG], *Kb*PETase exhibited comparable activity at 50 °C, achieving approximately half of the optimal catalytic activity of LCC[ICCG] (Supplementary Fig. 12)[16]. These results suggest that *Kb*PETase could serve as a more promising template for enzyme engineering compared to other WT enzymes, such as *Is*PETase and LCC, which exhibit extreme mesophilic or thermophilic traits.

## Sequence and structure analysis of *Kb*PETase

To provide structural insights into the enhanced PET-degrading activity of *Kb*PETase, we determined its structure using X-ray crystallography at a resolution of 1.75 Å (PDB: 9IW9). Overall, the structure of *Kb*PETase exhibits a high degree of conservation compared to *Is*PETase (PDB: 5XH3) (Fig. 4a). The catalytic triad (S128-H206-D176) and substrate pocket of *Kb*PETase are also conserved relative to those of *Is*PETase (S131-H208-D177) (Fig. 4b and c). This conservation suggests that VenusMine not only retains the overall structure scaffold but also preserves the active site, ensuring the enzymatic activity necessary for PET degradation.

A phylogenetic tree was constructed to include APET candidates alongside known PETases (Fig. 4d). The discovered enzymes are broadly distributed throughout the tree. One major clade contains most known PETases, including *Is*PETase, LCC, Cut190, and Thc_cut1,

**Table 1 | Kinetic parameters of *Kb*PETase, LCC and FastPETase derived from the Michaelis–Menten experiments**

|  | $K_m$ (mM) | $k_{cat}$ (S$^{-1}$) | $k_{cat}/K_m$ (mM$^{-1}\cdot$S$^{-1}$) |
|---|---|---|---|
| *Kb*PETase | 1.04 ± 0.201 | 0.270 ± 0.021 | 0.263 ± 0.031 |
| **FastPETase** | 1.57 ± 0.416 | 0.215 ± 0.029 | 0.141 ± 0.02 |
| **LCC** | 1.053 ± 0.289 | 0.186 ± 0.021 | 0.208 ± 0.017 |

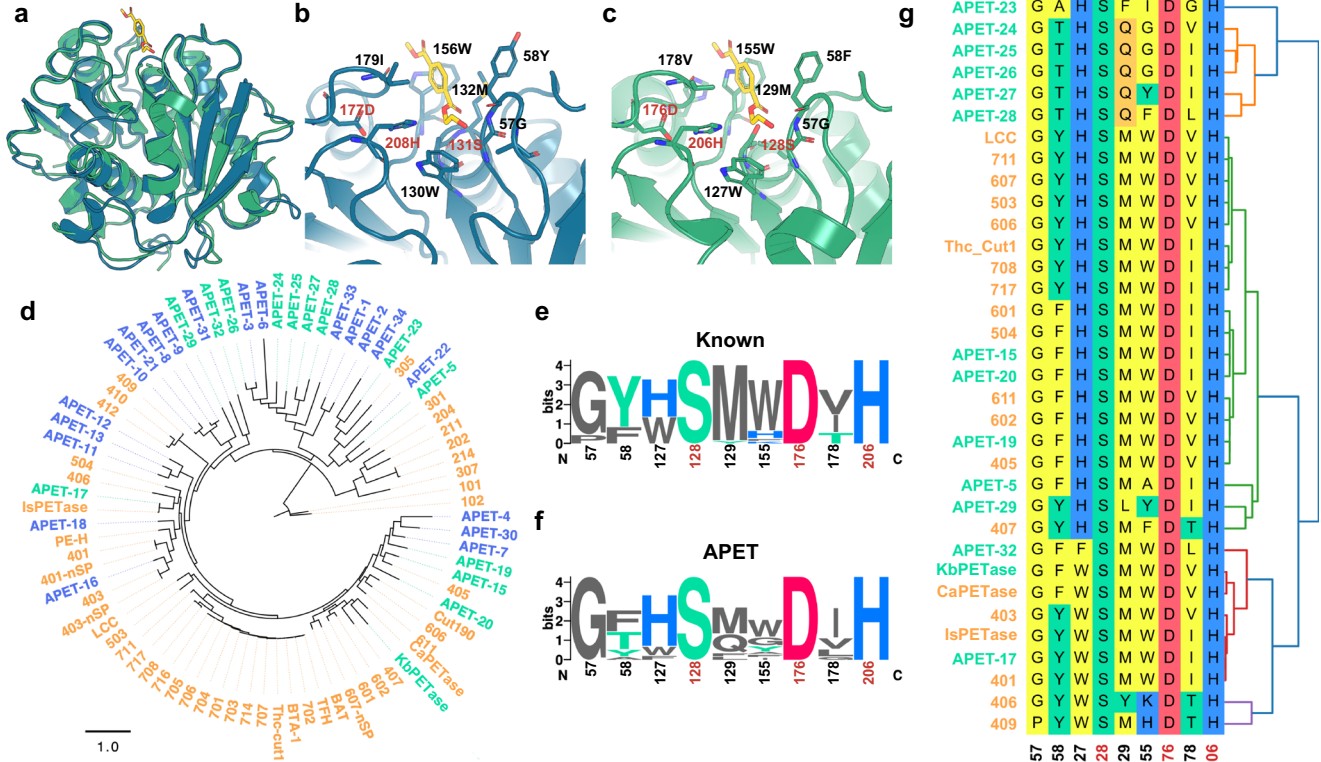

**Fig. 4 | Structure and sequence characteristics of APET enzymes and known PETases. a** Crystal structure of *Kb*PETase (green, PDB: 9IW9) and *Is*PETase (blue, PDB: 5XH3). The substrate structure (yellow) is derived from the *Is*PETase structure and structurally aligned with the *Kb*PETase pocket. Pocket structure of (**b**) *Is*PETase and (**c**) *Kb*PETase. **d** Phylogenetic tree of the 34 candidate proteins selected for experimental validation (blue and green) alongside previously reported PETases (orange). The APET candidates with confirmed activity are marked with green,

otherwise in blue. (**e**, **f**) Sequence logo plots for the pocket residues. The height of each letter indicates the conservation score for each site, indexed by the residue position in *Kb*PETase, with catalytic traid marked in red. **g** Structure-based alignment of pocket residues from APET (green) and known PETases (orange). Residues are annotated by their biochemical properties, with hierarchical clustering presented on the right. Residue indices correspond to their positions in *Kb*PETase.

from which *Kb*PETase also originates. Furthermore, several new clades emerged, populated by APET candidates positioned closer to the root of the phylogenetic tree. This arrangement suggests that VenusMine may effectively identify ancestral enzymes based on conserved structural features.

To further investigate the substrate pocket, structure alignment was performed using TM-align. Compared with reported PETases, the newly discovered APETs exhibit greater diversity in pocket residues while retaining a conserved catalytic triad (Fig. 4e–g). Notably, residues at positions 58, 129, and 155 show higher variability than those of known PETases (Fig. 4e, f). Hierarchical clustering based on pocket residues revealed that APETs are distributed across all major types of known PETases.

### Molecular dynamics analysis of *Kb*PETase

To elucidate the molecular basis of *Kb*PETase's enhanced thermostability, we performed comparative all-atom molecular dynamics (MD) simulations with LCC and *Is*PETase, representing thermophilic and mesophilic PET hydrolases, respectively. Analysis of the radius of gyration (Rg) distributions (Fig. 5a–c) revealed that *Kb*PETase predominantly adopts conformations with significantly reduced Rg values compared to LCC and *Is*PETase. This structural compactness correlates with its experimental thermostability, suggesting that reduced conformational entropy minimizes thermal unfolding propensity-a hallmark of enzymes adapted to fluctuating environmental conditions.

Root mean square fluctuation (RMSF) profiles further highlighted critical differences in local flexibility (Fig. 5d–f). While LCC and *Is*PETase exhibited pronounced dynamics in loop regions proximal to their catalytic pockets (highlighted as yellow sticks), *Kb*PETase demonstrated restricted flexibility in these functionally critical regions, with elevated fluctuations confined to peripheral N-terminal loops. This dichotomy suggests that *Kb*PETase achieves a balance between global stability and local flexibility, potentially optimizing both substrate accessibility and catalytic efficiency across a broad temperature range.

Analysis of non-covalent interactions revealed hierarchical trends in intramolecular stabilization (Fig. 5g–i). LCC, the most thermostable enzyme, maintained the highest global counts of salt bridges and hydrogen bonds, followed by *Kb*PETase and *Is*PETase. This gradient mirrors their experimental $T_m$, underscoring the role of cumulative non-bonded interactions in conferring thermostability. To further identify the critical hydrogen bonds in the catalytic pocket of the three proteins, we calculated the lifetime of every individual hydrogen bond within active sites (Details can be found in the section of Method). Notably, hydrogen bond lifetimes within the catalytic pocket—a metric reflecting bond persistence—showed *Kb*PETase surpassing both counterparts, with the N175-D205 interaction exhibiting the longest lifetime ($\tau = 5.00 \pm 0.21$ ns, Supplementary Fig. 13). Such prolonged hydrogen bonding may stabilize the catalytic triad geometry, facilitating substrate orientation and transition-state stabilization.

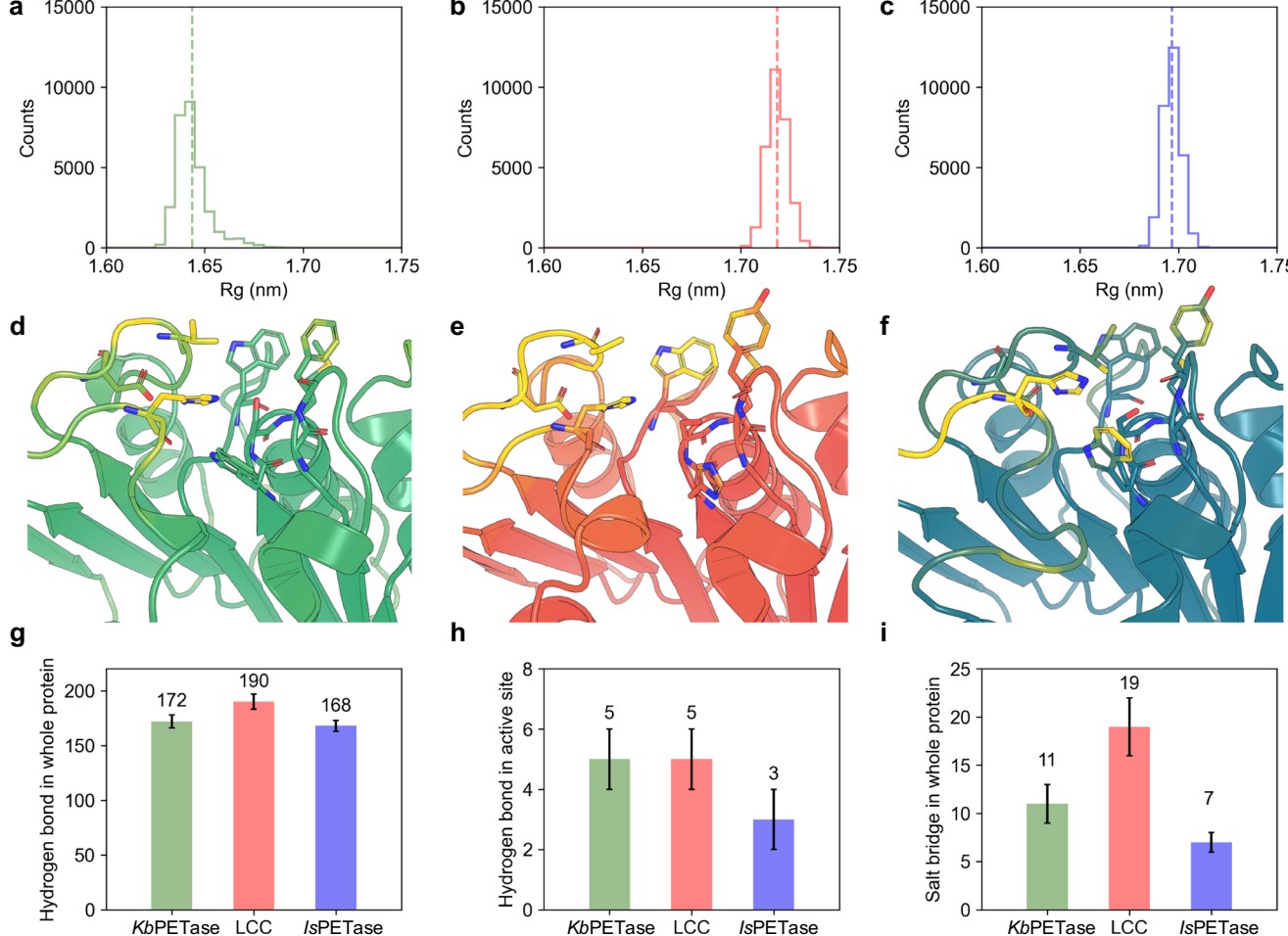

**Fig. 5 | Molecular dynamics simulation analyses of *Kb*PETase, *Is*PETase, and LCC. a–c** The distribution of Rg for *Kb*PETase (**a**), LCC (**b**) and *Is*PETase (**c**). **d–f** Residue-wise root mean square fluctuation (RMSF) mapped onto structures, with green (*Kb*PETase), red (LCC) and blue (*Is*PETase) for low RMSF and yellow for high RMSF. Residues in catalytic pockets are presented with sticks. **g–i** non-covalent interactions, including numbers of (**g**) global hydrogen bonds, (**h**) catalytic pocket hydrogen bonds, and (**i**) global salt bridges. Data are presented as mean values ± SD.

These findings collectively delineate the biophysical signature of *Kb*PETase: a compact scaffold resistant to entropic unfolding, paired with dynamic yet stable active site interactions. This dual strategy—global rigidity tempered by localized flexibility—may explain its unique functional profile, balancing thermostability with catalytic efficiency. By circumventing the traditional trade-off between thermal resilience and ambient-temperature activity observed in extremophilic enzymes, *Kb*PETase emerges as a versatile candidate for PET depolymerization processes requiring operational adaptability.

## Discussion

The increasing environmental impact of plastic waste, particularly PET, underscores the urgent need for innovative and effective biodegradation strategies[5,8,11,28]. Consequently, the search for promising PET-degrading enzymes has been widely pursued, leading to the successful characterization of several proteins to date[12,18–21,23]. However, current enzyme discovery methods predominantly rely on sequence-based analysis and are constrained by the high costs of experimental biochemical or biophysical protein structure characterization, as well as the limited accuracy of traditional computational folding simulations[60,61]. This reliance limits PETase discovery to a narrow sequence landscape[50].

Here, we present VenusMine, a structure-guided pipeline integrating FoldSeek-based structural searches, ProstT5-driven clustering, and deep learning-aided thermostability/solubility predictions. This methodology expands enzyme mining beyond sequence homology, enabling the discovery of *Kb*PETase. *Kb*PETase demonstrates remarkable catalytic performance, exhibiting superior activity at 50 °C, which establishes it as a highly practical candidate for industrial PET recycling. Structural and molecular dynamics analyses reveal a conserved catalytic architecture and strengthened intramolecular interactions, underpinning its enhanced thermostability and functional resilience. In contrast to engineered variants like FastPETase, *Kb*PETase's wild-type origin and unoptimized catalytic efficiency present a promising starting point for directed evolution, offering substantial potential for further refinement and application.

VenusMine not only expands the repertoire of known PETases but also establishes a robust framework for future enzyme discovery. With advancements in deep learning and protein language models, it may soon be possible to predict protein structures, and even complexes, more accurately, enabling the preliminary selection of candidate proteins through modeling. This approach not only aids in identifying exceptional enzyme molecules akin to "hidden gems" within databases but also enhances the accuracy and precision of the screening process. The emergence of *Kb*PETase provides an improved template for the engineering of high-performance enzymes, expanding the molecular library and offering new tools for the bioconversion and recycling of PET.

## Methods

### Structure search

In the structural search step, we utilized the default setting of the FoldSeek server website (https://search.foldseek.com/search). The structure of *Is*PETase (PDB: 5XFY) is selected as the only query protein for the FoldSeek search. Target structure database include AlphaFold databases (UniProt50 v4, SwissProt v4, Proteome v4), MGnify-ESM30 (v1), PDB100, and GMGCL 2204. The core structure is defined by a manual structure check of *Is*PETase, ranging from residue 51 to 234, covering the reported catalytic triad and the scaffold of the PET hydrolase. Any searched protein align range not covering this region was excluded from the following steps. This structure search step identified 4235 structural similar protein sequences. The Code of this part is available at step 2 in https://github.com/ai4protein/VenusMine.

### Sequence search

In the sequence search step, the sequences found in the structure search from different databases were merged and used as queries for the sequence search through the NCBI's NR database. This step is using MMseqs2 easy-search method with 2 rounds of iterations and sets max-seq to 2500, resulting in 33,247,501 sequences. After obtaining this vast repertoire, MMseqs2 easy-cluster method with 50% sequence identity is used to saving the cost for the following ProstT5 embeddings computation step. Resulting 436,488 representative proteins derived from 50% sequence identity clusters. Code for this part is available at steps 3 and 4 in https://github.com/ai4protein/VenusMine.

### Benchmarking protein language models for protein discovery

We systematically evaluated various computational approaches to assess their effectiveness in identifying and classifying proteins within enzyme commission (EC) groups, in order to find the optimal and most efficient method to discover novel enzymes. Utilizing a curated dataset of 265,488 protein sequences from the reviewed UniProt database (SwissProt), we focused on proteins up to 1000 amino acids in length and EC numbers with no less than 50 annotated enzymes. Our benchmarking compared traditional methods, including BLASTp and MMseqs2 for sequence similarity and FoldSeek with AlphaFold predictions for structural alignment against protein language models such as ESM-2 650 M, ESM-1b, and ProstT5. For each EC number, 10 starting points were randomly selected from the candidate pool, while other enzymes with annotated EC number as positive examples, enzymes without target EC number serve as negative examples. By calculating E-value for BLASTp, MMseqs2 and FoldSeek, and representation distances for protein language models, we are able to analyze the area under the ROC curve (AUROC) and area under precision-recall curve (AUPR) scores for 745 EC groups. This comprehensive evaluation provided insights into the strengths and limitations of each approach, highlighting the potential of protein language models in enhancing protein discovery and classification tasks. Results showed that both the structure-based methods FoldSeek and ProstT5 performed better than BLASTp methods, and ProstT5 is the only unsupervised deep-learning method that performed better than BLASTp (Supplementary Figs. 14–17). As a result, we select the structure-aware language model ProstT5 for our further studies.

### Clustering and representation tree built by structure-aware protein language model

We use the protein sequence-to-structure language model ProstT5 to calculate the embeddings of the protein candidates' sequences. Since ProstT5 is a bi-directional translation model between protein sequence and structure, the "AA2fold" mode of ProstT5 model is used to get a more structure-informed representation based on input sequences. Based on the representation of multiple sequences, we used agglomerative clustering to build a dendrogram for the sequences. The distances between representations are calculated as the Euclidean distance, and the agglomerative clustering method is using ward algorithm from scipy.cluster.hierarchy package. The code of this part is available at steps 5 and 6 in https://github.com/ai4protein/VenusMine.

### Protein expression and solubility prediction

Within the 2 selected cluster in representation tree, there are 6763 sequences advanced to the next screening step. We used a fine-tuned version of the ESM[47] to predict the melting temperature ($T_m$) of the candidates using datasets from the literature[48]. In addition, we employed ProtSolM to predict protein solubility[49]. Only candidates with predicted $T_m$ and solubility values exceeding those of the query sequence were retained, narrowing the pool to 223 sequences.

### Protein structure prediction and selection for APET

For further refinement, the structures of the 223 sequences were predicted using AlphaFold2[62], and any sequence with a pLDDT score below 75 was discarded. Structural alignment was then performed to

ensure that the candidate proteins possessed the catalytic active site necessary for PET hydrolase activity. Finally, the candidates were ranked by their predicted $T_m$ values, and the top 34 sequences were selected for wet-lab validation.

## Plasmid construction

The genes encoding APET1-34 were synthesized and optimized for expression (Supplementary Data 1) in *Escherichia coli* by Sangon Biotech (Shanghai, China). Signal peptides for APET1-34 were predicted using SignalP and subsequently removed from the synthetic DNA sequences. The synthesized genes for APET1-34 were then cloned into the *Nde*I and *Not*I sites of the pET-28a (+) expression vector, which features an N-terminal His-tag for protein purification. Detailed nucleotide sequences of APET1-34 are provided in Supplementary Data 1.

## Protein expression and purification

The expression plasmid was transformed into *Escherichia coli* BL21(DE3) competent cells. A 30 mL seed culture was grown at 37 °C in LB medium containing 50 μg/mL kanamycin, and then transferred to a 500 mL shaker flask containing the same antibiotic concentration. The cultures were incubated at 37 °C until the OD600 reached 1.0, at which point protein expression was induced by the addition of isopropyl-β-D-thiogalactopyranoside (IPTG) to a final concentration of 0.8 mM. Induction was followed by incubation at 16 °C for 16–20 h. Cells were harvested by centrifugation at $3345 \times g$ for 30 min, and the resulting pellets were collected for subsequent purification. The cell pellets were resuspended in lysis buffer (25 mM Tris-HCl, 500 mM NaCl, pH 7.4) and disrupted by ultrasonication (Scientz, China). The lysates were then centrifuged at $17418 \times g$ for 30 minutes at 4 °C, and the supernatants were loaded onto Ni-NTA columns (Smart Lifesciences, China, Ni NTA Beads 6FF: SA005500) pre-equilibrated with lysis buffer (25 mM Tris-HCl, 500 mM NaCl, pH 7.4). Following sample loading and subsequent washing steps, bound proteins were eluted using an elution buffer (25 mM Tris-HCl, 500 mM NaCl, 250 mM imidazole, pH 7.4). The protein was concentrated and buffer-exchanged using Amicon Ultra centrifugal filters (Millipore, 10 kDa MWCO) through repeated cycles of dilution with lysis buffer (25 mM Tris-HCl, 500 mM NaCl, pH 7.4) and centrifugation at $3345 \times g$ for 25 min intervals at 4 °C for ultrafiltration. Finally, the fractions containing the purified protein were flash-frozen at −20 °C in storage buffer (25 mM Tris-HCl, pH 7.4, 500 mM NaCl, 10% glycerol).

## Differential Scanning Fluorimetry (DSF)

The $T_m$ values were determined using the Differential Scanning Fluorimetry (DSF) method with the Protein Thermal Shift Dye Kit (Thermo Fisher, U.S.A.). To prepare the reaction mixture, 1.0 μL of SYPRO Orange Dye (SUPELCO, U.S.A) was diluted in 49 μL of lysis buffer (25 mM Tris-HCl, 500 mM NaCl, pH 7.4). Next, 1 μL of the diluted dye was combined with 19 μL of protein solution at a concentration of 0.1 mg/mL. DSF experiments were conducted using the LightCycler 480 Instrument II (ROChe, U.S.A). The reaction mixture was first equilibrated at 25 °C, then gradually heated to 99 °C at a rate of 0.05 °C/s, with a 2 min hold at the final temperature. Data processing was performed using the Protein Thermal Shift software.

## Nano differential scanning calorimetry (nanoDSC)

Nano DSC measurements were performed by using Nano DSC instruments (TA, U.S.A.). The concentration of PETases was 0.5 mg/ml in a buffer containing 25 mM Tris−HCl (pH = 7.5) and 500 mM NaCl. All the experiments were carried out at temperatures ranging from 10 to 110 °C with a heating rate of 1 °C/min and under a pressure of 3 atm. The melting curves of PETases were subtracted from the buffer scans.

## Enzymatic activity screening using *p*-Nitrophenyl Butyrate (*p*NPB) as substrate

Screening for activity on 4-Nitrophenol butyrate (*p*NPB). All reactions were conducted in 96-well plates with a total reaction volume of 100 μL. The final concentration of *p*NPB was 0.8 mM, and the enzyme concentration was 100 μg/mL. Each reaction mixture contained 10 μL of *p*NPB solution (dissolved in anhydrous ethanol), 80 μL of 10 mM potassium phosphate buffer (pH 8.0), and 10 μL of enzyme solution, which were incubated at 37 °C for 10 min. The reaction was terminated by adding 100 μL of anhydrous ethanol to quench the enzymatic activity. Each experiment was performed in triplicate. The concentration of p-nitrophenol was measured at 410 nm using an MD SpectraMax iD5 microplate reader (Molecular Devices, U.S.A.). The specific activity was calculated by determining the product quantity using a standard curve and then combining it with the enzyme concentration and reaction time. (Supplementary Fig. 4). All experiments were conducted in triplicate to ensure reproducibility.

For Kinetics analysis, enzymatic reactions involving 4-nitrophenyl butyrate (*p*NPB) were conducted in 96-well plates with a total reaction volume of 100 μL. The final *p*NPB concentrations ranged from 0.2 to 1.8 mM, while the enzyme concentration was maintained at 0.5 μg/mL. Each reaction mixture contained 10 μL of *p*NPB solution (dissolved in anhydrous ethanol), 80 μL of 10 mM potassium phosphate buffer (pH 8.0), and 10 μL of enzyme preparation, which was incubated at 50 °C for 3 min. The methods for reaction termination and detection were identical to those described in the preceding enzymatic activity screening section. One unit of enzyme activity was defined as the amount of enzyme required to convert 1 μmol of *p*NPB per minute. Kinetic parameters were determined by nonlinear regression analysis of the Michaelis-Menten equation using GraphPad Prism software (version 8.0), with initial velocity data obtained from triplicate measurements at varying substrate concentrations.

## PET depolymerization assay using amorphous PET film as substrate

In the PET degradation activity assays of 26 candidate proteins and the comparative analysis of *Kb*PETase with other wild-type enzymes, the following conditions were employed: amorphous PET film (Goodfellow, England) was cut into circular discs with a diameter of 6 mm for each reaction. The discs were incubated in 2900 μL of glycine-NaOH buffer (pH 9.0, 50 mM) containing 100 μL of enzyme solution (stock concentration 0.5 mg/mL) at 30, 40, 50, 55, 60 and 65 °C for 72 h. The reaction was terminated by heating the mixture at 90 °C for 10 min. Each sample was diluted to fall within the linear detection range for terephthalic acid (TPA) and mono(2-hydroxyethyl) terephthalate (MHET). After filtration through a 0.22 μm filter, the assay solution was analyzed by UPLC. All experiments were performed in triplicate. For the comparison of activity with LCC, LCC^ICCG, and FastPETase, the reaction buffers for all three enzymes were replaced with 100 mM potassium phosphate buffer, pH 8.0, while other conditions remained consistent with those described above.

## UPLC analysis

UPLC analysis was conducted using a Waters ACQUITY Arc system (Waters, U.S.A.) equipped with an autosampler and a UV detector set to 260 nm. The separation was performed on a Kinetex XB-C18 100 Å, 5 μm, 50 × 2.1 mm LC column (Phenomenex, U.S.A.) using a stepped, isocratic solvent gradient. Mobile phase A consisted of water with 0.1% formic acid, and mobile phase B was acetonitrile, with a fixed flow rate of 1.0 mL/min. Samples were injected at either 1 μL. Following injection, the mobile phase was maintained at 13% buffer B for 52 s to separate mono(2-hydroxyethyl) terephthalate (MHET) and terephthalic acid (TPA), then ramped up to 95% buffer B

for 33 seconds to separate larger reaction products and contaminants. The buffer was then returned to 13% for column re-equilibration, with a total run time of 1.8 min. Peaks were identified by comparison to chemical standards prepared from commercial TPA and in-house synthesized MHET, and the peak areas were integrated using software. Under these conditions, TPA eluted around 1.0 min, MHET around 2.3 min, and small amounts of bis(2-hydroxyethyl) terephthalate (BHET) and longer oligomers eluted between 2.7 and 3.2 min. The concentrations of TPA and MHET were determined by constructing standard curves.

### Crystallization and structure determination of *Kb*PETase

Crystals of *Kb*PETase, grown using hanging drop vapor diffusion by mixing equal volumes of protein and a buffer solution containing 1.26 M Sodium phosphate monobasic monohydrate,0.14 M Potassium phosphate dibasic at a temperature 291 K. Crystals were rapidly soaked in the reservoir solution supplemented with 20% glycerol as cryoprotectant, mounted on loops, and flash-cooled at 100 K in a nitrogen gas cryo-stream. Crystals Diffraction data was collected from a single crystal at Shanghai Synchrotron Radiation Facility (SSRF) BL18U beamline, China, with a wavelength of 1.75 Å at 100 K. The diffraction data were processed and scaled with HKL-3000. Relevant statistics were summarized in Supplementary Table 1.

The structure was solved by the molecular replacement method with a starting model predicted by AlphaFold II. The Initial model was build using PHENIX.autobuild. Manual adjustment of the model was carried out using the program COOT, and the models were refined by PHENIX.refinement and Refmac5. Stereochemical quality of the structures was checked by using PROCHECK. All of the residues locate in the favored and allowed region and none in the disallowed region. Refinement resulted in a model with excellent refinement statistics and geometry (Supplementary Table 1). The structure of *Kb*PETase was deposited in the Protein Data Bank, with PDB code 9IW9.

### Molecular dynamics simulations

The structures of proteins were obtained from the PDB database. For *Kb*PETase, we utilized the crystal structure resolved in this study, while PDB(8CMV) was used for the simulation of LCC and PDB (6EQG) for *Is*PETase. All of the three PDB are apo structure, without substrate. Protein and a large number of water molecules were filled in a cubic box. Chlorine counter ions were added to keep the system neutral in charge. The CHARMM27 force field was used for the complex and the CHARMM-modified TIP3P model was chosen for water[63–65]. The simulations were carried out at 310 K. After the 4000-step energy minimization procedure, the systems were heated and equilibrated for 100 ps in the NVT ensemble and 500 ps in the NPT ensemble. For each protein, 500 ns production simulations were conducted with a trajectory saving frequency of 10 ps. The final 300 ns (30,000 frames) were extracted for subsequent analysis. The integration step was set to 2 fs, and only the covalent bonds with hydrogen atoms were constrained by the LINCS algorithm. Lennard-Jones interactions were truncated at 12 Å with a force-switching function from 10 to 12 Å. The electrostatic interactions were calculated using the particle mesh Ewald method with a cutoff of 12 Å on an approximately 1 Å grid with a fourth-order spline. The temperature and pressure of the system were controlled by the velocity rescaling thermostat and the Parrinello-Rahman algorithm, respectively. All MD simulations were performed using the GROMACS 2025.1 software package.

### Lifetime of hydrogen bonds

To systematically evaluate the stability of hydrogen bonds within the catalytic sites of the three proteins, we computed the lifetime (τ) of each identified hydrogen bond using the following methodology:

First, the autocorrelation function of each hydrogen bond was calculated by:

$$C_x(t) = \frac{\langle h_{i,j}(t_0) \cdot h_{i,j}(t_0 + t) \rangle}{\langle h_{i,j}(t_0)^2 \rangle}$$

Here, $h_{i,j}(t_0)$ is a binary indicator (0 or 1) for hydrogen bond formation between donor $i$ and acceptor $j$ at time $t_0$, and the angle brackets denote time averaging over all trajectory frames. Subsequently, the lifetime (τ) of each hydrogen bond was derived by integrating the autocorrelation function over the total simulation time $T$:

$$\tau_x = \int_0^T C_x(t) dt$$

Here, $dt$ represents the trajectory sampling interval (10 ps). Larger τ values correspond to more stable hydrogen bonds, reflecting their persistence across the simulation.

### Reporting summary

Further information on research design is available in the Nature Portfolio Reporting Summary linked to this article.

## Data availability

The refined model of *Kb*PETase generated in this study has been deposited in the Protein Data Bank with PDB code 9IW9. All data that support the findings of this study are presented within the article and its supplementary files. For further inquiries or requests for additional information, please contact the corresponding authors. The raw data for the figures generated in this study are provided in the Source Data file. The detailed data of all of the PETases discovered can be found in the supplementary data. Source data are provided in this paper.

## Code availability

The code of the PETase discovery pipeline can be found in https://github.com/ai4protein/VenusMine. They can also be accessed via Zenodo in https://doi.org/10.5281/zenodo.15680583.

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

## Acknowledgements

This work was supported by Shanghai Municipal Science and Technology Major Project, the National Key Research and Development Program of China (2024YFA0917603), the Computational Biology Key Program of Shanghai Science and Technology Commission (23JS1400600), Shanghai Municipal Education Commission (2024AIZD015), Shanghai Jiao Tong University Scientific and Technological Innovation Funds (21X010200843), and Science and Technology Innovation Key R&D Program of Chongqing (CSTB2022TIAD-STX0017, CSTB2024TIAD-STX0032), the Student Innovation Center at Shanghai Jiao Tong University, and Shanghai Artificial Intelligence Laboratory. We would like to thank Teacher Jing Wang from Instrumental Analysis Center of Shanghai Jiao Tong University for assistance with PET hydrolysis performance tests via UPLC. Part of the computations in this paper were run on the Siyuan-1 cluster supported by the Center for High Performance Computing at Shanghai Jiao Tong University.

## Author contributions

L.Z., P.T., and L.H. conceptualized and supervised this research project. B.Z. and P.T. developed the methodology and designed the pipeline. B.Z., S.J., and M.L. implemented the method and conducted the in-silico screening of PETase. B.W. and R.H. conducted the wet-lab experiments. L.Z. and P.T. conducted MD simulations. B.W., L.Z., P.T., B.Z., and L.H. wrote the manuscript. All authors reviewed and accepted the manuscript.

## Competing interests

A patent application CN202410267798.X relating to the PETase discovered in this study has been filed in the name of Shanghai Jiao Tong University, pending. B.W. B.Z. P.T., and L.H. are the inventors of this patent. The other authors declare no competing interests.
