## [Transparent Peer Review file · Nature Communications]

Harnessing Protein Language Model for Structure-Based Discovery of Highly Efficient and Robust PET Hydrolases

Corresponding Author: Dr Lirong Zheng

Version 0:

Reviewer comments:

Reviewer #1

(Remarks to the Author)

Wu et al. used ProMine, which integrates the sequence-based and structure-based retrieval tools, to identify PET hydrolases (PETases). The authors utilize IsPETase as a template to explore and cluster proteins with potential PET-degrading activity, subsequently validating their function through biochemical assays. This method has identified 34 proteins, with 14 exhibiting PET degradation activity across a temperature range of 30-60°C.

1. A major concern is the recent publication of similar work in Science (DOI: 10.1126/science.adp5637), which demonstrated a similar study and introduced innovative results. Their innovation centered on the use of in silico techniques to identify clusters of PETase enzymes, successfully discovering novel PETases that outperform natural enzymes. Unfortunately, this manuscript undertakes alternative in silico method but yields significantly inferior results. While the similarity in research focus is common in competitive fields, the significantly enhanced catalytic effects of Kim's work substantially diminish the novelty of this work. Consequently, the contribution of this manuscript is relatively limited.

2. The manuscript details experiments with 34 enzymes but does not provide clear reasons for their selection. Is there expert experience guiding this choice, or are there specific criteria for selecting these enzymes? The authors should clarify their rationale for selecting these enzymes to enhance the scientific robustness of their study.

3. Some enzymes were assessed using pNPB for activity testing before switching to PET film. This choice is questionable. As it is well-known, enzymes that facilitate small molecule catalysis do not necessarily perform similarly with polymers like PET because active sites conducive to small molecule interactions are often too narrow for effective polymer binding. Hence, using pNPB as a substrate might not provide standardizable results. Authors should standardize on using PET film for enzyme activity tests instead of switching substrates without justified reasons.

4. When quantifying enzyme activity, the authors chose to use peak area measurement, which is perplexing especially since more precise and established methods have been detailed in the 2020 Carbios article in Nature (DOI: 10.1038/s41586-020-2149-4). The selection of non-standard measurement methods, particularly after enzyme screening, suggests a lack of enzymological depth in the authors' approach. Thus, it is strongly recommended that the authors adopt more accurate and standardized quantification methods for enzyme activity.

5. While KbPETase is highlighted for its performance, the comparison to established benchmarks like IsPETase and LCC appears inconsistent. For a meaningful benchmarking, it is essential to assess all enzymes under identical conditions and using the same substrates. The manuscript should address these discrepancies to provide a fair comparison of enzyme efficacy. For instance, IsPETase deactivates within one day at high temperatures, yet the authors extended the assay period to 72 hours. This inconsistency could compromise the reliability of the results. To ensure fairness in the comparison of enzyme activity, it is essential that measurements are taken within the linear range of enzyme activity, and specific activity should be quantified using units such as mgTPA h⁻¹ mgenzyme⁻¹. Moreover, when benchmarking against LCC, comparisons should be made at its optimal temperature of approximately 65°C, not 60°C. The buffer used throughout the experiments is not suitable for enzymes like LCC. A low-concentration K-Pi buffer, should be used instead. Also, when comparing with FastPETase, the authors should ensure that PET film is used as the substrate, not pNPB.

6. The manuscript only compares the activity of natural enzymes, which, while scientifically valid, lacks direct relevance to industrial applications where engineered enzymes, such as those from ICCG, exhibit significantly higher efficiencies. It's crucial for the study to extend its comparisons to these high-performance engineered enzymes under industrial- relevant conditions to enhance its industrial applicability.

In summary, the major issue with this manuscript is the lack of standardized experiments and direct comparisons under uniform conditions, which are critical for assessing the true innovative and practical value of the findings. The choice of inconsistent substrates and non-standardized quantification methods further complicates the scientific reliability of the

results. These factors, combined with the limited novelty, lead me to conclude that this manuscript is currently not suitable for publication in this journal or any other.

(Remarks on code availability)

Reviewer #2

(Remarks to the Author)

This study identified novel proteins capable of degrading polyethylene terephthalate (PET), specifically PET hydrolases (PETases). To achieve this, a pipeline called ProMine was applied, combining protein language models with structural similarity assessments to discover new PETase candidates. This approach led to the identification of 34 PETase candidates, of which 26 were successfully expressed and purified, while 14 demonstrated ester bond-cleaving activity. Among these, KbPETase from *Kibdelosporangium banguiense* distinguished itself with higher PET degradation activity at temperatures below 60 °C compared to other well-known PETases, such as LCC and TFH. However, despite its high melting temperature of 78.2 °C, KbPETase exhibited a significant decline in catalytic efficiency at 60 °C. The crystal structure of KbPETase was determined and structurally analyzed, including molecular dynamics (MD) simulations.

The study is thorough, the research is well-conducted, and the manuscript is clearly written. With the novel results obtained, publication could be feasible after satisfactorily addressing the following issues:

- 1) I. 62: It should be noted that PHL-7 is also known as PES-H1 (ACS Catal 12, 9790–9800 (2022)). Later on in the manuscript, the authors mention the PETase PE-H. Do they mean PHL-7/PES-H1 with this?
- 2) I. 69: In JACS Au 2024, 4, 10, 4000–4012 it was analyzed in detail why/how PET-PET interactions have to be overcome in order for PET to be able to enter the active site. This paper should thus be cited.
- 3) Fig. 2: For completeness, the results of IsPETase should be added to panel b). Can the authors speculate on a correlation between the results shown in panels b), c) and d), i.e. can one make predictions for PET degradation efficiency based on the efficiency to hydrolyze pNPB and T_m?
- 4) Fig. 3: Why was in panels c) and d) the comparison made to FastPETase while the latter was not used in the comparisons before. Please add FastPETase to panels a) and b) as well as add the LCC results to panels d) and d).
- 5) The MD simulations conducted in this study are relatively short, and the observed instability in the results for IsPETase warrants a more detailed analysis. It is important to specify the structure used for these simulations, including the PDB code, and to confirm whether this structure was resolved in complex with a substrate. Additionally, the authors need to investigate the implications of the high RMSF values encountered during the simulation of IsPETase.

To strengthen the findings, I recommend including LCC in the simulations and either extending the simulation duration to 500 ns or running them in triplicate for a total of 300 ns (3 x 100 ns). Furthermore, the greater number of hydrogen bonds in the catalytic site of KbPETase may significantly contribute to its enhanced stability and catalytic efficiency compared to other PETases. Therefore, a detailed analysis of these hydrogen bonds is necessary, including identification of the residues involved and the stability of these bonds throughout the simulations.

(Remarks on code availability)

- 1) The link <https://github.com/Zuricho/Enzyme-Discovery-Codebase> does not work.
- 2) The link <https://github.com/Zuricho/ProMine> given in the paper works. However, I could only find the License there, but no code itself.

On the other hand, ProMine only uses existing code and provides the pipeline for using these codes; therefore, there is no need to review the code here.

Reviewer #3

(Remarks to the Author)

This manuscript by Wu *et al.* presents a study on the discovery of polyethylene terephthalate (PET) esterases for application in biodegradation/recycling processes. A protein discovery algorithm, ProMine, was developed to identify PETases based on structural similarities followed by *in silico* characterizations of solubility and thermostability. This led to the selection of 34 proteins which, based on preliminary biochemical characterization, ultimately led to the identification of a PETase from the actinomycete *Kibdelosporangium banguiense*. In addition to its biochemical properties, the crystal structure of this enzyme as determined.

Given their potential to help with the recycling of plastics, PETases have attracted considerable attention. A number of successful studies on PETases involving a combination of rational engineering, directed evolution, and advanced computational strategies have been reported that identified/developed PETases with enhanced properties, including

increased enzyme melting temperature. Indeed, the crystal structure of at least 12 PETases from a variety of microbes (from psychrophiles to thermophiles) have been determined and studied, as well as the generation of a synthetic enzyme. In the current study, the authors developed a search pipeline that led to their study of enzyme that had already been identified as a family member. Moreover, it is not clear that the properties of this enzyme are better (as claimed) than others that have been investigated/generated. For example, whereas the authors claim thermostability at 60 degrees C, the KbPETase was considerably less so than others when assayed for PET film degradation as illustrated in Figure 3a. This, and a number of other issues would need to be addressed.

1. The manuscript in general should be shortened. For example, lines 85–108 should be replaced with a simple statement of purpose; currently this text represents an abstract detailing the findings of the study. Similarly, the Discussion reiterates most of the results in detail; this needs to be condensed.

2. There are issues with the kinetic analyses. It would appear that an end-point assay was used. As analyses of steady-state parameters require data manipulation of initial velocity, the authors need to know (and demonstrate) that such was obtained under the conditions employed for assays. Thus, they need to show representative time curves of activity with the enzyme and substrate concentrations used that the 10 min end point of their routine assays sits within the linear portion of the curve to reflect initial velocity (this is particularly important given the different enzyme concentrations used throughout - see below).

Also, different reporting of activity with respect to units was used, some of which are incorrect, and none appear to use the definition of activity as presented in the Methods. For example, Figure 2b reports “enzyme activity” as “ $\mu\text{m}/\text{nmol}\cdot\text{min}$ ” – are the authors presenting Specific Activity and mean to report μmoles (of product) per nmol PETase $\cdot\text{min}$? With Figure 3c reporting the degradation of film, V (S^{-1}) is used as the measure of activity. However, the units of S^{-1} indicate the constant k_{cat} which can only be derived from a steady-state analysis of enzyme kinetics. Moreover, as a constant it does not change with substrate concentration.

Table 1 presents kinetic parameters using up to up to 6 significant figures, and no indication of error. The error should be noted, and reflecting the accuracy of the experiment, probably no more than three significant figures.

Finally, regarding the kinetic data presented in Table 1, the difference between the parameters of Kb and Fast PETase would be recognized by kineticists as being trivial and so the text of Lines 239 – 240 should be edited accordingly (a 1.4 increase is recognized as insignificant).

3. Line 590: The crystal structure of KbPETase was determined and the PDB accession number of 9IW9 is reported. However, neither 9IW9 nor 91W9 return the structure – “No items found.” Also, a search of the PDB using *Kibdelosporangium banguiense* and PETase did not return it. If the structure has yet to be released by the PDB, then the wwPDB Validation Report should be provided in Supplementary Information.

Minor issues:

Line 171: If the aim of the study was to identify thermostable enzymes, why use an assay temperature of 37 degrees C (i.e., that of the human body)?

Figure 2a could be deleted - at least moved to supplementary information.

Lines 491 and 494: g force and not rpm should be stated for centrifugation (rpm are meaningless without the identity of the rotor used).

Line 495: Presumably, the equilibration/loading buffer for the Ni-NTA chromatography was the same as the lysis buffer but this should be stated.

Line 497: the conditions used for ultracentrifugation should be stated.

Line 497-8: How were the fractions assayed for protein content?

Lines 560-571: To avoid redundancy given the enzyme assays used were the same, the description of the kinetic analyses should be integrated with the Lines 519-530 describing the screening for activity.

Line 571: was non-linear regression used for the determination of kinetic parameters?

Also:

Line 525, was it established that a final concentration of 18% ethanol is sufficient to quench activity?

Lines 521 and 563: Is it correct as stated that the final concentrations of the PETase was 100 $\mu\text{g}/\text{ml}$ was the screening (line 521) and 0.5 $\mu\text{g}/\text{ml}$ for the kinetic analysis (line 563). If so, then the question about initial velocity (above) is even more important given the 200 fold difference in concentrations.

(Remarks on code availability)

Version 1:

Reviewer comments:

Reviewer #2

(Remarks to the Author)

The authors very carefully addressed all comments. I have no further requests.

(Remarks on code availability)

Reviewer #3

(Remarks to the Author)

My issues with the original manuscript have been addressed adequately in the revised version.

(Remarks on code availability)

REVIEWER COMMENTS

Response to Comments - Reviewer 1

Wu et al. used ProMine, which integrates the sequence-based and structure-based retrieval tools, to identify PET hydrolases (PETases). The authors utilize IsPETase as a template to explore and cluster proteins with potential PET-degrading activity, subsequently validating their function through biochemical assays. This method has identified 34 proteins, with 14 exhibiting PET degradation activity across a temperature range of 30-60 °C.

Reply 1: We thank the reviewer for highlighting our key findings.

1. A major concern is the recent publication of similar work in Science (DOI: 10.1126/science.adp5637), which demonstrated a similar study and introduced innovative results. Their innovation centered on the use of in silico techniques to identify clusters of PETase enzymes, successfully discovering novel PETases that outperform natural enzymes. Unfortunately, this manuscript undertakes alternative in silico method but yields significantly inferior results. While the similarity in research focus is common in competitive fields, the significantly enhanced catalytic effects of Kim's work substantially diminish the novelty of this work. Consequently, the contribution of this manuscript is relatively limited.

Reply 2: While both our study and Kim et al. (DOI: 10.1126/science.adp5637) focus on identifying novel wild-type PETases, our methodologies are fundamentally different. Our study and Kim's research were conducted independently and did not have the opportunity for cross-referencing. Kim et al. used a traditional sequence-based approach to discover high-performance wild-type enzymes, such as MipaPETase and KubuPETase. In contrast, our study developed a structure-based screening pipeline guided by structure-aware protein language models (PLMs), enabling more refined and efficient identification of structurally relevant PETase candidates.

Our method introduces two key innovations:

1. Structure-guided expansion of search space:

Unlike conventional structure-based approaches that rely solely on tools like FoldSeek or DALI to align against precomputed structure databases (typically constrained by the prediction coverage of AlphaFold), our method incorporates a sequence-based expansion step following structure-guided querying. This hybrid strategy allows a more comprehensive search across the vast sequence landscape for proteins with structural similarity to known PETases. As shown in Fig. R1 (Fig. 1d in the revised main text), the identified candidates in our approach covers most of the validated PETases, thereby broadening the discovery space of PET hydrolyases. From the results, this outcome is challenging to achieve through sequence alignment alone.

2. Deep-learning-based candidate filtering:

We utilize structure-informed embeddings from protein language models (ProstT5) alongside deep learning-based predictions of thermostability (ESM2) and solubility (ProtSolM), enabling more

accurate functional characterization and prioritization. This filtering step enhances the biochemical validation success rate by focusing on structurally plausible and biochemically viable candidates.

Fig. R1. In silico structure-based PETase discovery pipeline and clustering results. A representation tree constructed using embeddings from ProstT5. Green bars: proteins with validated PETase activity; blue bars: proteins with the correct EC number; red bars: proteins lacking correct EC number. Two clusters (dark red and orange) are selected for further experimental screening.

In terms of functional performance, *KbPETase*—identified through our pipeline—demonstrates superior catalytic activity under identical experimental conditions. Specifically, *KbPETase* shows a 1.64-fold higher PET degradation activity at 50 °C compared to *KubuPETase* at 60 °C, and a 2.24-fold greater activity than *MipaPETase* at 40 °C (Fig. R2). These results suggest that, despite differences in discovery strategies, *KbPETase* outperforms previously reported PETases in catalytic efficiency across a range of temperature conditions.

Fig. R2. PET film degradation activity of *KbPETase* compared with *KubuPETase* and *MipaPETase*. Reaction was conducted in 50 mM Glycine-NaOH (pH 9.0) across various temperatures (30 °C, 40 °C, 50 °C, 55 °C, 60 °C, and 65 °C) for 72 h.

2. The manuscript details experiments with 34 enzymes but does not provide clear reasons for their selection. Is there expert experience guiding this choice, or are there specific criteria for selecting these enzymes? The authors should clarify their rationale for selecting these enzymes to enhance the scientific robustness of their study.

Reply 3: In the revised main text, we have added an explanation of the enzyme selection criteria (Fig. R3). The process is as follows:

1. Language model clustering and initial selection:

Following ProST5-based clustering, we prioritized Cluster 1 and Cluster 3 for candidate selection. These two clusters contain all major reported high-activity wild-type PETases (e.g., *IsPETase*, LCC). A full list of known PETases in these clusters is provided in Supplementary Table 3.

2. Thermostability and solubility screening: a. T_m prediction: All sequences in the selected clusters were evaluated using a fine-tuned ESM2 model to predict their T_m . **b. Solubility filtering:** ProtSolM was employed to predict solubility. Only sequences with a predicted T_m higher than that of *IsPETase* ($T_m = 45.8$ °C) and high solubility scores were retained for further analysis.

3. Structural validation: a. ESMFold-predicted structures of the filtered candidates were structurally aligned with *IsPETase* (PDB: 5XFY) using TM-align. Candidates with TM-score below 0.5 were excluded. **b.** Proteins lacking the conserved PETase catalytic triad (D177-H208-S131 in *IsPETase*) were also excluded from further consideration.

4. Final selection: The remaining candidates were grouped by sequence similarity and ranked by predicted T_m in descending order. Due to practical constraints of wet-lab validation, we selected the top 34 candidates with the highest predicted T_m values. One previously reported protein (PDB: 5YFE) is removed.

Fig R3. Selection pipeline for discovered PET hydrolases.

3. Some enzymes were assessed using *p*NPB for activity testing before switching to PET film. This choice is questionable. As it is well-known, enzymes that facilitate small molecule catalysis do not necessarily perform similarly with polymers like PET because active sites conducive to small molecule interactions are often too narrow for effective polymer binding. Hence, using *p*NPB as a substrate might not provide standardizable results. Authors should standardize on using PET film for enzyme activity tests instead of switching substrates without justified reasons.

Reply 4: The use of *p*NPB is a well-established method in PET hydrolase research for preliminary screening of ester bond hydrolysis activity, due to the ease and efficiency of *p*NPB hydrolysis assays. This approach has been widely employed in the discovery and characterization of many PETases (Science 2016, 351, 1196–1199; Angew. Chem. Int. Ed. 2022, 61, e202203061; Appl. Environ. Microbiol. 2012, 78, 1556–1562; Appl. Microbiol. Biotechnol. 2014, 98, 10053–10064; Macromolecules 2011, 44, 4632–4640).

We fully acknowledge that *p*NPB is not a direct analog of PET, and its results do not directly translate to polymer degradation capacity. For this reason, in our study, *p*NPB was used solely as a preliminary screen to identify proteins with general ester bond hydrolytic activity (Fig. 2a in the revised main text). All subsequent evaluations of PET degradation performance were conducted using PET film as the substrate under standardized reaction conditions (Fig. 3a in the revised main text). This two-step strategy-rapid screening with *p*NPB followed by detailed analysis using PET film-is aligned with established protocols in the field (e.g., Science 2016, 351, 1196–1199; Angew. Chem. Int. Ed. 2022, 61, e202203061).

4. When quantifying enzyme activity, the authors chose to use peak area measurement, which is perplexing especially since more precise and established methods have been detailed in the 2020 Carbios article in Nature (DOI: 10.1038/s41586-020-2149-4). The selection of non-standard measurement methods, particularly after enzyme screening, suggests a lack of enzymological depth

in the authors' approach. Thus, it is strongly recommended that the authors adopt more accurate and standardized quantification methods for enzyme activity.

Reply 5: Follow the reviewer's suggestion, we have re-analyzed the data. The results demonstrated that the activity of APET-14 (*KbPETase*) is 97-fold and 156-fold higher than that of *IsPETase* at 30 °C and 50 °C, respectively (Fig. R4, Fig. 2a in the revised main text). After precise quantification, the degradation activity of the protein was found to be comparable to that obtained using the aforementioned methods.

Fig. R4. Biochemical characterization of selected PETase candidates. PET film degradation assays were conducted in 50 mM Glycine-NaOH buffer (pH 9.0) at multiple temperatures (30 °C, 40 °C, 50 °C, 55 °C, and 60 °C) for 72 hours. Reactions were performed in triplicate; data are presented as mean values \pm SD.

5. While *KbPETase* is highlighted for its performance, the comparison to established benchmarks like *IsPETase* and *LCC* appears inconsistent. For a meaningful benchmarking, it is essential to assess all enzymes under identical conditions and using the same substrates. The manuscript should address these discrepancies to provide a fair comparison of enzyme efficacy. For instance, *IsPETase* deactivates within one day at high temperatures, yet the authors extended the assay period to 72 hours. This inconsistency could compromise the reliability of the results. To ensure fairness in the comparison of enzyme activity, it is essential that measurements are taken within the linear range of enzyme activity, and specific activity should be quantified using units such as mgTPA h⁻¹ mgenzyme⁻¹.

Reply 6: We conducted time-course analyses for both *IsPETase* and *KbPETase* under their respective optimal conditions. As shown in Fig. R5, *IsPETase* exhibited a plateau in activity after 24 hours at 30 °C, consistent with its known instability. In contrast, *KbPETase* maintained linear product accumulation over a 72-hour period at 50 °C, demonstrating superior thermostability and

sustained catalytic activity. Extended incubation times of 48-96 hours or even 168 hours are commonly used in PETase studies for evaluating enzyme performance, particularly when assessing cumulative degradation capability. This methodology is employed in many studies (Nat. Commun. 2022, 13, 7850; Angew. Chem. Int. Ed. 2022, 61, e202203061; Science 2025, 387(6729): eadp5637), where enzyme activity is typically reported based on the total degradation products yields (e.g., TPA, MHET) rather than time-normalized specific activity. To ensure fairness, all enzymes in our study—including *Is*PETase, LCC, and *Kb*PETase were initially evaluated under identical reaction conditions (50 mM Glycine-NaOH buffer, pH 9.0, for 72 hours) across a temperature gradient spanning 30 °C, 40 °C, 50 °C, 55 °C, 60°C, and 65 °C (Fig. 3a in the revised main text).

Regarding the use of specific activity units (e.g., mg TPA h⁻¹ mg enzyme⁻¹), we agree that this format offers a standardized basis for comparison. However, this characterization overlooks the other degradation product, MHET. The use of total product release is a commonly adopted approach in reported PETase studies. To ensure consistency with the reporting conventions of foundational PETase studies (Science 2016, 351, 1196–1199; Nature 2022, 604, 662–667; Nat. Catal. 2022, 5, 673–681; Science 2025, 387(6729): eadp5637), we chose to report enzymatic performance based on total degradation product yields.

Fig. R5. Time-course analysis of *Is*PETase and *Kb*PETase PET degradation activity. Reactions were conducted in 50 mM Glycine-NaOH (pH 9.0) using PET films at the respective optimal temperatures (30 °C for *Is*PETase and 50 °C for *Kb*PETase), with samples collected at 24, 48, and 72 hours. All reactions were performed in triplicate; data are presented as mean ± SD.

Moreover, when benchmarking against LCC, comparisons should be made at its optimal temperature of approximately 65 °C, not 60 °C. The buffer used throughout the experiments is not suitable for enzymes like LCC. A low-concentration K-Pi buffer, should be used instead. Also, when

comparing with FastPETase, the authors should ensure that PET film is used as the substrate, not pNPB.

Reply 7: Following the reviewer's suggestions, we have supplemented our tests, which did not qualitatively alter the conclusions. The result shows that *KbPETase* exhibited 1.47-fold higher activity at 50 °C than LCC at 65 °C (Fig. R6, Figure 3a in the revised main text).

Fig. R6. Activity measurement of discovered PETases. PET film degradation activity of the previously characterized PETases compared to *KbPETase*. The reaction was conducted in 50 mM Glycine-NaOH (pH 9.0) at various temperatures (30 °C, 40 °C, 50 °C, 55 °C, 60 °C and 65 °C) for 72 h. The reactions of LCC and *KbPETase* were respectively conducted in K-Pi buffer (pH 8.0) at 65 °C and 50 mM Glycine-NaOH (pH 9.0) at 50 °C for 72 h, respectively. Reactions were performed in triplicate; data are presented as mean values \pm SD.

Some studies also employ Glycine buffer as a standardized condition for comparing the activity of wild-type enzymes with LCC (Nat Commun. 2023;14(1):4556). However, we have supplemented our analysis with tests under each enzyme's optimal conditions. We measured the PET degradation activity of both enzymes under their respective optimal catalytic conditions for 72h: *KbPETase* at 50 °C in 50 mM Glycine-NaOH buffer (pH 9.0) and LCC at 65°C in K-Pi buffer (pH 8.0). Notably, the finding that *KbPETase* exhibits higher activity than LCC remains unchanged. Under these matched conditions, *KbPETase* demonstrated 17% higher activity than LCC (Fig. R7, Fig. S10 in the revised SI), emphasizing its robustness across varying assay setups and its competitive catalytic efficiency.

In response to the reviewer's concern about FastPETase, we confirm that all comparisons involving FastPETase were performed using PET film as the substrate, not pNPB. A detailed analysis comparing the activity of *KbPETase* and FastPETase across a range of temperatures is provided in Fig. R8 (Fig. S11 in the revised SI).

Fig. R7. Direct comparison of *KbpPETase* and LCC in optimal catalytic conditions. The PET film degradation activity was measured after 72 h for *KbpPETase* at 50 °C in 50 mM Glycine-NaOH buffer (pH 9.0) and for LCC at 65 °C in K-Pi buffer (pH 8.0). Reactions were performed in triplicate; results are shown as mean ± SD.

Fig. R8. Comparison of PET film degradation activity between *KbpPETase* and FastPETase. Reactions were carried out at 30 °C, 40 °C, 50 °C, and 60 °C in 50 mM Glycine-NaOH (pH 9.0) for *KbpPETase* and in K-Pi buffer (pH 8.0) (Nature 2022, 604, 662–667) for FastPETase. All reactions were conducted for 72 h in triplicate; data are shown as mean ± SD.

6. The manuscript only compares the activity of natural enzymes, which, while scientifically valid, lacks direct relevance to industrial applications where engineered enzymes, such as those from ICCG, exhibit significantly higher efficiencies. It's crucial for the study to extend its comparisons to these high-performance engineered enzymes under industrial- relevant conditions to enhance its industrial applicability.

Reply 8: Follow reviewer's suggestion, we did a direct comparison between *KbpPETase* and LCC-ICCG (Fig. R9, Fig. S 12 in the revised SI), the latter being a heavily engineered PETase with four residues mutated (Nature 2020, 580, 216–219). As can be seen, the best activity of *KbpPETase*

discovered in the present work appears at 50 °C, and it is about half of the best activity of ICCG occurring at 65°C. This is expected as *KbPETase* is a natural wildtype enzyme without any mutational engineering, whose performance cannot compete with a heavily engineered mutant. A meaningful comparison should be made either between wild type proteins as done in the present work, or between heavily mutated enzymes (Nat. Commun. 2023, 14, 4556; Science 2025, 387(6729): eadp5637; ACS Catal. 2023;13(20):13156-13166). We respectfully note that comparing the performance of our wild-type enzyme to extensively engineered variants may not be appropriate, as the latter have undergone rounds of artificial optimization to enhance their catalytic properties. This is analogous to comparing the developmental capabilities of a one-month-old infant with those of a fully matured adult—the comparison reflects differences in the extent of engineering rather than intrinsic potential. We note that, *KbPETase* as a wild-type protein has outstanding activity compared to all other wildtype ones, and can be a good starting point for further mutational engineering for industrial applications. However, such engineering work requires a large amount of extra work and exceeds the scope of the present work. The focus of the present work is to provide an LLM-based protein-minding method, and to use PETase as an example to demonstrate this new method can easily find a pool of catalytically potent natural scaffolds that can serve as promising starting points for future enzyme engineering.

Fig. R9. Activity comparison of *KbPETase* and ICCG. PET film degradation activity of ICCG and *KbPETase* across a temperature range (30 °C to 65 °C). Reactions were performed in K-Pi buffer (pH 8.0) for ICCG and 50 mM Glycine-NaOH (pH 9.0) for *KbPETase*, incubated for 72 hours. All reactions were conducted in triplicate; data are shown as mean \pm standard deviation.

In summary, the major issue with this manuscript is the lack of standardized experiments and direct comparisons under uniform conditions, which are critical for assessing the true innovative and practical value of the findings. The choice of inconsistent substrates and non-standardized

quantification methods further complicates the scientific reliability of the results. These factors, combined with the limited novelty, lead me to conclude that this manuscript is currently not suitable for publication in this journal or any other.

Reply 9: We would like to address the concerns raised as follows:

1. Methodological Innovation and Independent Study: Our study represents a methodological advancement compared to Kim et al (DOI: 10.1126/science.adp5637). Our study is driven by AI and integrates PLMs to achieve structure-based discovery of novel PETases. Compared to traditional sequence alignment methods, our approach significantly expands the search space and incorporates functional screening capabilities, thereby enhancing screening efficiency. This has enabled us to identify novel PET-degrading enzymes, including the high-performance wild-type PET hydrolase, *KbPETase*.

2. Evaluation System and Standardized Testing: Regarding the evaluation system, we respectfully argue that our methodology adheres to standard practices in the field. In the initial screening process of candidate molecules, we employed *p*NPB to evaluate ester bond cleavage activity. This approach has been widely employed in the discovery and characterization of many PETases (Science 2016, 351, 1196–1199; Angew. Chem. Int. Ed. 2022, 61, e202203061; Appl. Environ. Microbiol. 2012, 78, 1556–1562; Appl. Microbiol. Biotechnol. 2014, 98, 10053–10064; Macromolecules 2011, 44, 4632–4640). All subsequent evaluations of PET degradation performance were conducted using PET film as the substrate under standardized reaction condition (Nat. Commun. 2022, 13, 7850; Angew. Chem. Int. Ed. 2022, 61, e202203061; Science 2016, 351, 1196–1199; Nature 2022, 604, 662–667; Nat. Catal. 2022, 5, 673–681; Science 2025, 387(6729): eadp5637).

3. Supplementary Tests and Consistency of Findings: We have carefully implemented the additional testing methods suggested by the reviewer. Importantly, these supplementary tests did not qualitatively alter our conclusions. *KbPETase* consistently demonstrated higher degradation activity compared to LCC and *IsPETase*, while also exhibiting excellent stability, making it a strong candidate for further exploration.

In summary, our study introduces a new, structure-based screening pipeline that complements traditional methods, adheres to standardized testing practices, and yields reliable and reproducible results. We believe these points address the reviewer's concerns while highlighting the unique contributions of our work.

Response to Comments - Reviewer 2

*This study identified novel proteins capable of degrading polyethylene terephthalate (PET), specifically PET hydrolases (PETases). To achieve this, a pipeline called ProMine was applied, combining protein language models with structural similarity assessments to discover new PETase candidates. This approach led to the identification of 34 PETase candidates, of which 26 were successfully expressed and purified, while 14 demonstrated ester bond-cleaving activity. Among these, KbPETase from *Kibdelosporangium banguiense* distinguished itself with higher PET degradation activity at temperatures below 60 °C compared to other well-known PETases, such as LCC and TFH. However, despite its high melting temperature of 78.2 °C, KbPETase exhibited a significant decline in catalytic efficiency at 60 °C. The crystal structure of KbPETase was determined and structurally analyzed, including molecular dynamics (MD) simulations.*

The study is thorough, the research is well-conducted, and the manuscript is clearly written. With the novel results obtained, publication could be feasible after satisfactorily addressing the following issues:

Reply 1: We thank the reviewer for highlighting our key findings.

Comments:

1. l. 62: It should be noted that PHL-7 is also known as PES-H1 (ACS Catal 12, 9790–9800 (2022)). Later on in the manuscript, the authors mention the PETase PE-H. Do they mean PHL-7/PES-H1 with this?

Reply 2: Based on Ref. (ACS Catal. 2022, 12, 9790–9800) and Ref. (Front. Microbiol. 2022, 11, 114), PE-H is a wild-type PETase derived from *Pseudomonas aestusnigri*, which is distinct from PHL-7 (also referred as PES-H1), a wild-type PETase identified from a compost metagenome (ChemSusChem 2022, 15, e202101062). In our manuscript, PE-H was used as one of the representative wild-type PETases for benchmarking PET degradation activity among natural enzymes. We have now cited the appropriate reference (Front. Microbiol. 2022, 11, 114) for PE-H in the Introduction section of the revised main text.

2. l. 69: In JACS Au 2024, 4, 10, 4000–4012 it was analyzed in detail why/how PET-PET interactions have to be overcome in order for PET to be able to enter the active site. This paper should thus be cited.

Reply 3: We have now cited this paper in the revised main text.

3. Fig. 2: For completeness, the results of IsPETase should be added to panel b). Can the authors speculate on a correlation between the results shown in panels b), c) and d), i.e. can one make predictions for PET degradation efficiency based on the efficiency to hydrolyze pNPB.

Reply 4: We tested the pNPB degradation activity of IsPETase and have included it in Fig. R10 (Fig. 2a in the revised main text).

Fig. R10. Experimental validation and biochemical characterization of PETase candidates. Enzyme activity of 14 PETase candidates showing effective ester bond hydrolysis, as measured using *p*-nitrophenyl butyrate (*p*NPB) as a substrate at 37 °C. Reactions were performed in triplicate and data are presented as mean values ± SD.

We investigated potential correlations among the data shown in panels b), c), and d). Specifically, we calculated the correlation between *p*NPB hydrolysis activity and melting temperature (T_m). The analysis revealed relatively low correlation coefficients (both Spearman and Pearson) among these features, suggesting that neither T_m nor *p*NPB activity can be used to reliably predict the other (Fig. R11, Fig. S6 in the revised SI). These findings highlight the complexity of PETase activity and suggest that multiple factors likely contribute to overall degradation performance. We have added above discussion and Fig. R11 in the revised main text and SI, respectively.

Fig. R11. Correlation between enzymatic activity and T_m for PETases shown in Fig. 2. *Is*PETase is highlighted in red, while the discovered APETase variants are shown in blue. A linear regression line (gray), along with Pearson (r) and Spearman (ρ) correlation coefficients, is included in the figure.

4. Fig. 3: Why was in panels c) and d) the comparison made to FastPETase while the latter was not used in the comparisons before. Please add FastPETase to panels a) and b) as well as add the LCC results to panels d) and d).

Reply 5: We have added the result of FastPETase in Fig. R8 (Fig. S11 in the revised SI) and the results of LCC in Fig. R12 and Table R1 (Fig. 3c and d, Table 1 in the revised main text).

Fig. R12. Activity measurements of *Kb*PETase, FastPETase, and LCC. (a) Comparison of the enzymatic kinetics curves of *Kb*PETase, LCC and FastPETase using *p*NPB as the substrate. Reactions were performed in triplicate and data are presented as mean values \pm SD. (b) The depolymerization percentages of PET film by *Kb*PETase and FastPETase at 50 °C, as well as by LCC at 65 °C, were determined through UPLC analysis of the released products. Note that the optimal enzymatic activity of *Kb*PETase, FastPETase, and LCC is 50 °C, 50 °C, and 65 °C, respectively. Reactions were conducted in triplicate, and the results are presented as mean values \pm SD.

Table R1. Kinetic parameters of *Kb*PETase, LCC and FastPETase derived from the Michaelis–Menten experiments (Fig. R12)

	K_m (mM)	K_{cat} (S^{-1})	K_{cat}/K_m ($\text{mM}^{-1}\cdot S^{-1}$)
Kb PETase	1.04 ± 0.201	0.270 ± 0.021	0.263 ± 0.031
FastPETase	1.57 ± 0.416	0.215 ± 0.029	0.141 ± 0.02

LCC

1.053 ± 0.289

0.186 ± 0.021

0.208 ± 0.017

5. The MD simulations conducted in this study are relatively short, and the observed instability in the results for *IsPETase* warrants a more detailed analysis. It is important to specify the structure used for these simulations, including the PDB code, and to confirm whether this structure was resolved in complex with a substrate. Additionally, the authors need to investigate the implications of the high RMSF values encountered during the simulation of *IsPETase*.

To strengthen the findings, I recommend including LCC in the simulations and either extending the simulation duration to 500 ns or running them in triplicate for a total of 300 ns (3 x 100 ns). Furthermore, the greater number of hydrogen bonds in the catalytic site of *KbPETase* may significantly contribute to its enhanced stability and catalytic efficiency compared to other PETases. Therefore, a detailed analysis of these hydrogen bonds is necessary, including identification of the residues involved and the stability of these bonds throughout the simulations.

Reply 6: In the revised main text, we have re-performed longer and more detailed MD simulations, including LCC as a new control for comparative analysis. Regarding the PDB structures used in the MD simulations:

- ***KbPETase*:** We utilized the crystal structure resolved in this study (PDB: 9IW9), which does not contain a substrate molecule.
- **LCC:** PDB structure 8CMV (unliganded).
- ***IsPETase*:** PDB structure 6EQG (unliganded).

For each protein, 500 ns simulations were conducted with a trajectory storage frequency of 10 ps. Results from the updated MD trajectories are shown in Fig. R13 (Fig. 5 in the revised main text). Analysis of the radius of gyration (R_g) distributions (Fig. R13a-c) revealed that *KbPETase* adopts conformations with reduced R_g values compared to LCC and *IsPETase*. This structural compactness suggests that reduced conformational entropy minimizes thermal unfolding propensity—a hallmark of enzymes adapted to fluctuating environmental conditions.

Root mean square fluctuation (RMSF) profiles also highlighted differences in local flexibility (Fig. R13d-f). While LCC and *IsPETase* exhibited pronounced dynamics in loop regions proximal to their catalytic pockets (highlighted as yellow sticks), *KbPETase* demonstrated restricted flexibility in these functionally critical regions, with elevated fluctuations confined to peripheral N-terminal loops. This dichotomy suggests that *KbPETase* achieves a balance between global stability and localized flexibility, potentially optimizing both substrate accessibility and catalytic efficiency across a broad temperature range.

Analysis of non-covalent interactions revealed hierarchical trends in intramolecular stabilization (Fig. R13g-i). LCC, the most thermostable enzyme, maintained the highest global counts of salt

bridges and hydrogen bonds, followed by *KbPETase* and *IsPETase*. This gradient mirrors their experimental T_m , underscoring the role of cumulative non-bonded interactions in conferring thermostability. To further identify the critical hydrogen bonds in the catalytic pocket of the three proteins, we calculated the lifetime of every individual hydrogen bond within active sites (Details can be found in the section of Method in the revised manuscript). Notably, hydrogen bond lifetimes within the catalytic pocket—a metric reflecting bond persistence—showed *KbPETase* surpassing both counterparts, with the N175-D205 interaction exhibiting the longest lifetime ($\tau = 5.00 \pm 0.21$ ns, Fig. R14, Supplementary Fig. 13). Such prolonged hydrogen bonding may stabilize the catalytic triad geometry, facilitating substrate orientation and transition-state stabilization.

The above discussion has been added into the main text.

Fig. R13. Molecular dynamics simulation analyses of *KbPETase*, *IsPETase*, and LCC. (a-c) The distribution of Rg for (a) *KbPETase*, (b) LCC, and (c) *IsPETase*. (d-f) Residue-wise root mean square fluctuation (RMSF) mapped onto structures, with green (*KbPETase*), red (LCC) and blue (*IsPETase*) for low RMSF and yellow for high RMSF. Residues in catalytic pockets are presented with sticks. Non-covalent interactions, including numbers of (g) global hydrogen bonds, (h) catalytic pocket hydrogen bonds, and (i) global salt bridges.

Fig. R14 Lifetime analysis of critical catalytic pocket hydrogen bonds. *KbpETase* (left) retains a stable N175-D205 hydrogen bond ($\tau = 5$ ns), exceeding lifetimes $\tau = 4.35$ ns in LCC (middle) and $\tau = 3.36$ ns *IsPETase* (right).

Remarks on code availability

- 1) The link <https://github.com/Zuricho/Enzyme-Discovery-Codebase> does not work.
- 2) The link <https://github.com/Zuricho/ProMine> given in the paper works. However, I could only find the License there, but no code itself. On the other hand, ProMine only uses existing code and provides the pipeline for using these codes; therefore, there is no need to review the code here.

Reply 7: We have now updated our code on GitHub, with the link: <https://github.com/Zuricho/VenusMine> (note the name has been changed).

Response to Comments - Reviewer 3

*This manuscript by Wu et al. presents a study on the discovery of polyethylene terephthalate (PET) esterases for application in biodegradation/recycling processes. A protein discovery algorithm, ProMine, was developed to identify PETases based on structural similarities followed by in silico characterizations of solubility and thermostability. This led to the selection of 34 proteins which, based on preliminary biochemical characterization, ultimately led to the identification of a PETase from the actinomycete *Kibdelosporangium banguense*. In addition to its biochemical properties, the crystal structure of this enzyme as determined.*

Reply 1: We thank the reviewer for highlighting our key findings.

Given their potential to help with the recycling of plastics, PETases have attracted considerable attention. A number of successful studies on PETases involving a combination of rational engineering, directed evolution, and advanced computational strategies have been reported that identified/developed PETases with enhanced properties, including increased enzyme melting temperature. Indeed, the crystal structure of at least 12 PETases from a variety of microbes (from psychrophiles to thermophiles) have been determined and studied, as well as the generation of a synthetic enzyme. In the current study, the authors developed a search pipeline that led to their study of enzyme that had already been identified as a family member. Moreover, it is not clear that the properties of this enzyme are better (as claimed) than others that have been investigated/generated. For example, whereas the authors claim thermostability at 60 degrees C, the KbPETase was considerably less so than others when assayed for PET film degradation as illustrated in Figure 3a. This, and a number of other issues would need to be addressed.

Reply 2: While several studies have successfully reported PETases with improved properties through rational engineering, directed evolution, or computational design, most discovery strategies still rely on either (i) isolation from culturable microorganisms (*Science* 2016, 351, 1196–1199; *Appl. Microbiol. Biotechnol.* 2014, 98, 10053–10064), or (ii) sequence-based approaches that identify candidates with high sequence similarity to known PETases (*Appl. Environ. Microbiol.* 2012, 78, 1556–1562; *Angew. Chem. Int. Ed.* 2022, 61, e202203061). These methods often miss distant homologs with low sequence similarity, do not incorporate structural information into the search, and typically screen only a small number of candidates (often fewer than 15) due to limited pre-selection capabilities (*Nat. Commun.* 2023, 14, 4556; *ChemSusChem* 2022, 15, e202101062).

Our approach advances the field in two key aspects:

1. Structure-guided mining with expanded search space: Unlike conventional methods that rely solely on predicted structure databases and alignment tools such as FoldSeek or DALI, our pipeline integrates structure-based search with a comprehensive sequence-based expansion step. This dual strategy leverages the rapidly growing sequence databases to identify proteins with structural similarity to known PETases, regardless of their sequence similarity. As a result, we significantly expanded the candidate pool, as shown in Fig. R1, and successfully recovered known PETases that lack prior annotation.
2. Deep learning-guided prioritization: We utilized protein language models (PLMs) and deep

learning tools for predicting thermostability, solubility, and functional relevance of candidate proteins. This allowed us to enrich the candidate set for functionally promising enzymes, resulting in a high hit rate during experimental validation.

Regarding the family of *KbPETase*, our discovery process did not depend on predefined protein family annotations. The phylogenetic tree (Fig. R15, also Fig. 4d in the revised main text) was constructed only after identifying *KbPETase*, our lead candidate. This tree revealed that *KbPETase* belongs to a clade containing most previously reported high-performance PETases, such as CaPETase, LCC, Cut190, and The_Cut1. We hypothesize that this clade represents an evolutionarily enriched group of potent PET-degrading enzymes. Notably, a recent study (Science 2025, 387(6729): eadp5637) has independently confirmed the high activity of KubuPETase, a member of this same clade, further validating its functional significance.

Fig. R15. Structure and sequence characteristics of APET enzymes and known PETases. Phylogenetic tree of the 34 candidate proteins selected for experimental validation (blue and green) alongside previously reported PETases (orange). The APET candidates with confirmed activity are marked with green, otherwise in blue.

For the performance of the protein, *KbPETase* demonstrates exceptionally high degradation activity, surpassing that of current wild-type proteins and even outperforming the engineered variant FastPETase. Its superior performance underscores its potential as a highly efficient enzyme for PET degradation. We evaluated its PET degradation activity and determined that its optimal catalytic temperature is 50 °C (Fig. R4; Fig. 2c in the revised main text). In particular, its activity at 50 °C surpasses *IsPETase* by 97-fold and exceeds LCC at 65°C by 47%, highlighting its potential as a robust and efficient candidate for PET degradation in practical settings (Fig. R16a and R16b below; Fig. 3a and 3b in the revised main text). Although the performance of *KbPETase* at 60 °C is lower

than that of LCC (a thermophilic enzyme optimized for high-temperature catalysis), it demonstrates outstanding activity at moderate temperatures, including 30°C and 50°C—temperatures more relevant to ambient or low-energy industrial conditions. It is important to note that PET degradation activity is generally benchmarked at each enzyme’s optimal catalytic temperature, as supported by previous studies (Science 351, 1196–1199, 2016; Nat. Commun. 14, 4556, 2023; Science 387(6729): eadp5637, 2025; Nat. Commun. 13, 7850, 2022; Nature 604, 662–667, 2022; Angew. Chem. Int. Ed. 62, e202218390, 2023; Nat. Catal. 5, 673–681, 2022). Comparisons made under non-optimal conditions (e.g., evaluating a mesophilic enzyme like *KbPETase* at 60 °C) do not fairly reflect enzymatic potential and may lead to misleading conclusions.

Fig. R16. Activity and thermostability measurements of discovered PETases. (a) PET film degradation activity of the previously characterized PETases compared to *KbPETase*. The reaction was conducted in 50 mM Glycine-NaOH (pH 9.0) at various temperatures (30 °C, 40 °C, 50 °C, 55 °C, 60 °C and 65 °C) for 72 h. Reactions were performed in triplicate; data are presented as mean values ± SD. (b) Comparison of the T_m of *KbPETase* with other reported WT PETases using DSF. Reactions were performed in triplicate; data are presented as mean values ± SD.

We note that while generative AI models, InstructPLM (bioRxiv 2024.04.17.589642) have been proposed for PETase design in non-peer-reviewed work, we found no verifiable activity data provided in these works. To date, no published studies have demonstrated synthetic enzymes rivaling natural or engineered benchmarks like LCC or FastPETase in catalytic performance. *Kb*PETase's wild-type origin and unoptimized catalytic efficiency (surpassing engineered variants like FastPETase in k_{cat}/K_M and TPA production) position it as a uniquely promising template for further engineering. We have revised the manuscript to explicitly contextualize these comparative advantages and clarify the methodological innovation of ProMine, which expands the repertoire of industrially relevant biocatalysts by bridging structural and functional diversity.

1. The manuscript in general should be shortened. For example, lines 85–108 should be replaced with a simple statement of purpose; currently this text represents an abstract detailing the findings of the study. Similarly, the Discussion reiterates most of the results in detail; this needs to be condensed

Reply 3: In response to these suggestions, we have streamlined the manuscript by condensing lines 85-108 into a focused statement of purpose and restructuring the Discussion.

2. There are issues with the kinetic analyses. It would appear that an end-point assay was used. As analyses of steady-state parameters require data manipulation of initial velocity, the authors need to know (and demonstrate) that such was obtained under the conditions employed for assays. Thus, they need to show representative time curves of activity with the enzyme and substrate concentrations used that the 10 min end point of their routine assays sits within the linear portion of the curve to reflect initial velocity (this is particularly important given the different enzyme concentrations used throughout - see below)

Reply 4: We reanalyzed the time-course data and confirmed that all enzymatic reactions proceeded linearly over the 10-minute assay period under the conditions used for kinetic measurements. Therefore, our endpoint data are well justified and reliable. Representative time-course curves have now been included (Fig. R17, below) to demonstrate that the 10-minute endpoint lies within the linear range of the reaction. These data confirm that the rates measured reflect initial velocities, thereby validating the reliability of our kinetic parameter calculations (Angew. Chem. Int. Ed. 61, e202203061, 2022; Science 351, 1196–1199, 2016).

Fig. R17. Kinetics Curve of pNPB Assay for Candidate PETases.

3. Also, different reporting of activity with respect to units was used, some of which are incorrect, and none appear to use the definition of activity as presented in the Methods. For example, Figure 2b reports “enzyme activity” as “ $\mu\text{m}/\text{nmol}\cdot\text{min}$ ” – are the authors presenting Specific Activity and mean to report μmoles (of product) per nmol PETase $\cdot\text{min}$?

Reply 5: Thank the referee. We have corrected the typo “ $\mu\text{m}/\text{nmol}\cdot\text{min}$ ” to specific activity unit ($\mu\text{mol}\cdot\text{nmol}^{-1}\cdot\text{min}^{-1}$), which is defined as the amount of product (in μmoles) generated per nmol of enzyme per minute ($\mu\text{mol product nmol}^{-1}\text{ enzyme min}^{-1}$). and consistent with our Methods section and shown in Figure R10.

4. With Figure 3c reporting the degradation of film, V (S^{-1}) is used as the measure of activity. However, the units of S^{-1} indicate the constant k_{cat} which can only be derived from a steady-state analysis of enzyme kinetics. Moreover, as a constant it does not change with substrate concentration.

Reply 6: Thank you for your valuable comment regarding the use of units in Figure 3c. Recognizing that V (S^{-1}) inappropriately suggested k_{cat} values, we have corrected Figure 3c (Figure R12a) to display activity in $\mu\text{mol}\cdot\text{min}^{-1}\cdot\text{mg}^{-1}$, accurately reflecting our experimental measurements of pNPB degradation. This revision enhances the clarity and precision of our data presentation.

5. Table 1 presents kinetic parameters using up to up to 6 significant figures, and no indication of error. The error should be noted, and reflecting the accuracy of the experiment, probably no more than three significant figures.

Reply 7: In response to this comment, we have revised Table 1 (and Table R1 below) by limiting kinetic parameters to three significant figures and including standard errors, ensuring accurate representation of experimental precision and alignment with best practices for kinetic data

presentation.

Table R1. Kinetic parameters of *Kb*PETase, LCC and FastPETase derived from the Michaelis–Menten experiments

	Km (mM)	Kcat (S-1)	Kcat/ Km (mM-1·S-1)
Kb PETase	1.04 ± 0.201	0.270 ± 0.021	0.263 ± 0.031
FastPETase	1.57 ± 0.416	0.215 ± 0.029	0.141 ± 0.02
LCC	1.053 ± 0.289	0.186 ± 0.021	0.208 ± 0.017

6. Finally, regarding the kinetic data presented in Table 1, the difference between the parameters of *Kb* and Fast PETase would be recognized by kineticists as being trivial and so the text of Lines 239 – 240 should be edited accordingly (a 1.4 increase is recognized as insignificant).

Reply 8: While FastPETase is a highly optimized, engineered variant, our data show that the wild-type *Kb*PETase exhibits comparable—if not slightly superior—performance in both PET degradation and thermostability (see Fig. R8). In response to Reviewer 2’s comment, we conducted additional kinetic analyses, which further support *Kb*PETase’s favorable catalytic efficiency and substrate affinity compared to FastPETase (Table R1). We agree that a 1.4-fold difference in *k*_{cat} may be considered modest from a strict kinetic standpoint. However, it is important to emphasize that *Kb*PETase achieves this level of performance as a wild-type enzyme, without requiring directed evolution or extensive mutagenesis. This highlights its potential as a valuable starting scaffold for future engineering efforts, which is highlighted in the main text.

3. Line 590: The crystal structure of *Kb*PETase was determined and the PDB accession number of 9IW9 is reported. However, neither 9IW9 nor 9IW9 return the structure – “No items found.” Also, a search of the PDB using *Kibdelosporangium banguense* and PETase did not return it. If the structure has yet to be released by the PDB, then the wwPDB Validation Report should be provided in Supplementary Information.

Reply 9: We thank the reviewer for raising this issue. The unavailability of *Kb*PETase structure results from our delayed release request to maintain confidentiality. To address this, we have included the official wwPDB validation report in the Supplementary Information, providing structural validation metrics.

Minor issues:

Line 171: If the aim of the study was to identify thermostable enzymes, why use an assay temperature of 37 degrees C (i.e., that of the human body)?

Reply 10: The *p*NPB screening served solely as an initial step to identify proteins with ester bond

cleavage activity for subsequent detailed characterization. Thermal stability was assessed through T_m measurements and PET film degradation activity, which provide more practical and application-relevant data (ACS Catal, 13(20):13156-13166, 2023.).

Figure 2a could be deleted - at least moved to supplementary information.

Reply 11: We have moved this figure to the Supplementary Information.

Lines 491 and 494: g force and not rpm should be stated for centrifugation (rpm are meaningless without the identity of the rotor used).

Reply 12: In response to this suggestion, we have standardized our methodology by replacing all rotation speed (rpm) values with relative centrifugal force (RCF) in $\times g$ units, including in lines 481 and 484, ensuring reproducibility across different centrifuge models

Line 495: Presumably, the equilibration/loading buffer for the Ni-NTA chromatography was the same as the lysis buffer but this should be stated.

Reply 13: The Ni-NTA chromatography equilibration/loading buffer was identical to the lysis buffer (25 mM Tris-HCl, 500 mM NaCl, pH 7.4). We have explicitly stated this in lines 485-486 to ensure protocol transparency and reproducibility.

Line 497: the conditions used for ultracentrifugation should be stated.

Reply 14: In response to this comment, we have detailed the ultracentrifugation conditions in lines 488-491: protein concentration and buffer exchange were performed using Amicon Ultra centrifugal filters (Millipore, 10 kDa MWCO) through repeated cycles of lysis buffer (25 mM Tris-HCl, 500 mM NaCl, pH 7.4) dilution and centrifugation at $3345\times g$ for 25 minutes at $4^\circ C$.

Line 497-8: How were the fractions assayed for protein content?

Reply 15: Following ultrafiltration, protein concentrations were determined using Nanodrop spectrophotometry (Thermo Scientific, U.S.A) with extinction coefficient corrections, while purity was verified by SDS-PAGE. Samples were then diluted with lysis buffer to a final concentration of 0.1 mg/mL based on measurements, ensuring experimental consistency and quantification accuracy.

Lines 560-571: To avoid redundancy given the enzyme assays used were the same, the description of the kinetic analyses should be integrated with the Lines 519-530 describing the screening for activity.

Reply 16: We have streamlined the Methods section by integrating kinetic analyses with activity screening in lines 514-538, creating a coherent workflow from initial screening to detailed characterization.

Line 571: was non-linear regression used for the determination of kinetic parameters?

Reply 17: We determined kinetic parameters through non-linear regression analysis using GraphPad

Prism (v8.0), fitting triplicate initial velocity measurements at varying substrate concentrations to the Michaelis-Menten equation for accurate K_m and V_{max} determination (lines 566-569).

Also:

Line 525, was it established that a final concentration of 18% ethanol is sufficient to quench activity?

Reply 18: As previously mentioned, the use of *p*NPB for assessing ester bond cleavage activity is a well-established method, with ethanol termination being a widely adopted practice, as demonstrated in recent studies (Nat. Commun 15, 1417, 2024).

Lines 521 and 563: Is it correct as stated that the final concentrations of the PETase was 100 $\mu\text{g/ml}$ was the screening (line 521) and 0.5 $\mu\text{g/ml}$ for the kinetic analysis (line 563). If so, then the question about initial velocity (above) is even more important given the 200 fold difference in concentrations.

Reply 19: Our 10-minute kinetic profiles demonstrate linearity, confirming method validity (Fig R8). The kinetic analysis follows established protocols (Nat. Commun 15, 1417, 2024. Angew. Chem. Int. Ed 61, e202203061, 2022.) to ensure reliability.